



# A new form of the Saint-Venant equations for variable topography

Cheng-Wei Yu[1], Ben R. Hodges[1], and Frank Liu[2]

[1]National Center for Infrastructure Modeling and Management, The University of Texas at Austin
[2]Oak Ridge National Laboratory

**Correspondence:** Ben R. Hodges (hodges@utexas.edu)

**Abstract.** The solution stability of river models using the one-dimensional (1D) Saint-Venant equations can be easily undermined when source terms in the discrete equations do not satisfy the Lipschitz smoothness condition for partial differential equations. Although instability issues have been previously noted, they are typically treated as model implementation issues rather than as underlying problems associated with the form of the governing equations. This study proposes a new "reference slope" form of the Saint-Venant equations to ensure smooth source terms and eliminate potential numerical oscillations. It is shown that a simple algebraic transformation of channel geometry provides a smooth reference slope while preserving the correct cross-sectional flow area and the total Piezometric pressure gradient that drives the flow. The reference slope method ensures the slope source term in the governing equations is Lipschitz-continuous while maintaining all the underlying complexity of the real-world geometry. The validity of the mathematical concept is demonstrated with the open-source SPRNT model in a series of artificial test cases and simulation of a small urban creek. Validation comparisons are made with analytical solutions and the HEC-RAS model. The new method reduces numerical oscillations and instabilities without requiring ad hoc smoothing algorithms.

## 1 Introduction

The Saint-Venant equations (SVE) for one-dimensional (1D) river modeling are typically presented with pressure forcing terms of either (i) gradients of the water surface elevation or (ii) thalweg bottom slope combined with gradients of the water depth. In this study we demonstrate a new form using a reference slope ($S_R$) and its associated depth ($h_a$), which are shown to be algebraically identical to the two standard forms of the SVE. The new forms provide greater flexibility in addressing numerical convergence issues associated with modeling discontinuous bottom slopes. A key point of this paper is that precise representation the thalweg bottom slope ($S_0$) and hydrostatic pressure gradients ($\partial h_0/\partial x$) is *not* necessary to correctly represent variable topography. Indeed, the splitting point for representing the forcing Piezometric pressure gradient as a body-force (defined by a slope) and a residual head gradient term is free choice in a simple algebraic substitution. Different choices for the splitting provide different body force directions and lead to different forms of the SVE – all of which are valid representations of





variable topography and do not constitute "smoothing" of topography. We will show that it is possible to use a smooth slope (body force) term in the SVE without actually smoothing the topography. Herein, this smooth slope term will be designated a "reference slope," $S_R$, to distinguish it from the traditional thalweg bottom slope, $S_0$.

The two common differential forms of the SVE, with the different terms highlighted in blue, are

$$\frac{\partial Q}{\partial t} + \frac{\partial}{\partial x}\left(\frac{Q^2}{A}\right) = -gAS_f - gA\frac{\partial \eta}{\partial x} \tag{1}$$

$$\frac{\partial Q}{\partial t} + \frac{\partial}{\partial x}\left(\frac{Q^2}{A}\right) = -gAS_f + gAS_0 - gA\frac{\partial h_0}{\partial x} \tag{2}$$

where $Q$ is the flow rate, $A$ is the cross-sectional area, $\eta$ is the water surface elevation, $h_0$ is the thalweg depth, $S_0$ is the thalweg bottom slope, and $S_f$ is the friction slope that represents the local energy gradient. Equation (1) can be envisioned as using the Piezometric head gradient to force the flow, as shown on the left-hand-side (LHS) of Fig. 1. In contrast, eq. (2) can be

envisioned as splitting the Piezometric gradient into a body force in the bottom slope direction and a hydrostatic head gradient, as shown on the right-hand-side (RHS) of Fig. 1. Note that both equations are valid for variable topography, despite only the second equation explicitly representing the bottom slope and thalweg-depth hydrostatic pressure gradients.

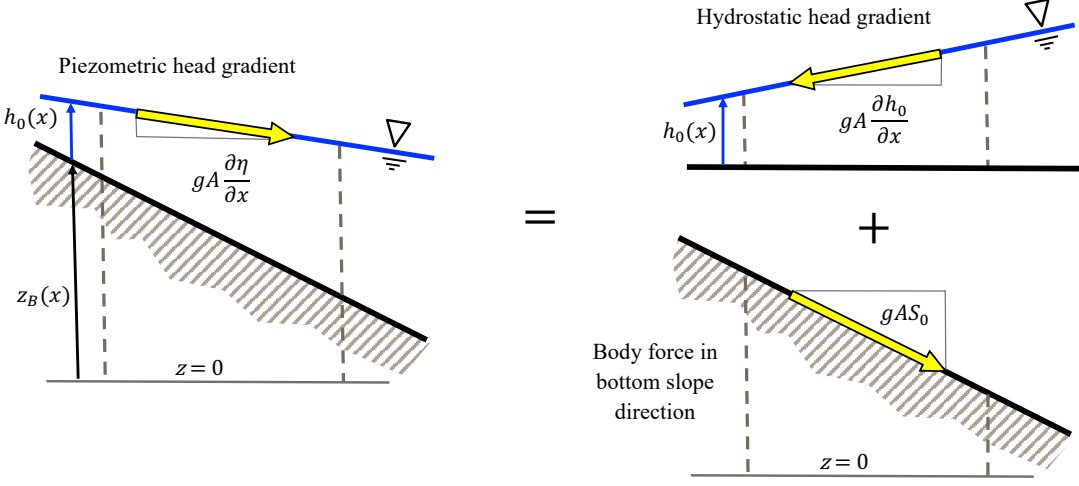

**Figure 1.** The $\eta$ form of SVE on the left has a driving Piezometric head gradient, which is equivalent (on the right) to the sum of the hydrostatic head gradient and a body force aligned with $S_0$. The effect of varying geometry is handled in $A$ in both forms.

The two SVE forms of eqs.(1) and (2) are algebraically identical using the identity

$$\frac{\partial \eta}{\partial x} \equiv \frac{\partial h_0}{\partial x} - S_0 \tag{3}$$





However, we can propose a more general identity of

$$\frac{\partial \eta}{\partial x} \equiv \frac{\partial h_a}{\partial x} - S_R \tag{4}$$

where $S_R$ is an arbitrary reference slope and $h_a$ is an associated depth consistent with the above definition. Applying the above to eq. (1) provides

$$\frac{\partial Q}{\partial t} + \frac{\partial}{\partial x}\left(\frac{Q^2}{A}\right) = -gAS_f + gAS_R - gA\frac{\partial h_a}{\partial x} \tag{5}$$

where the terms highlighted in blue are equivalent to those in eqs. (1) and (2). Clearly, if we let $S_R = S_0$ then $h_a = h_0$ and we recover eq. (2). Furthermore, if we let $S_R = 0$ then $h_a = \eta$ and we recover eq. (1). The equations are algebraically identical with these substitutions, so it follows that using a reference slope of zero ($S_R = 0$) must exactly represent the same topographic variability as using a reference slope that mimics the topographic slope ($S_R = S_0$), as long as the $h_a$ is correctly defined consistent with eq. (4). That is, from simple algebra the use of the real $S_0$ in the SVE *is not* required to capture effects

of topographic variability as long as the "depth" gradient term is correctly redefined as something other than the thalweg depth gradient.

From the arguments above, the effects of varying bottom topography are captured by $S_R = 0$ and $h_a = \eta$, which implies we are also free to introduce any other (preferably smooth) $S_R$ into eq. (5) without altering the representation of variable topography. An example is illustrated in Fig. 2. As the splitting defined in eq. (4) makes eq. (5) algebraically identical to

eqs. (1) and (2), the introduction of a smooth $S_R$ does not reflect "smoothing" of the topography. It is merely reflects a decision on whether effects of non-smoothness will reside solely in solution variables $A$ and $h_a$, or will also be forced as a non-smooth source term in $S_0$.

We would like to use an *a priori* smooth $S_R$ in a computational model rather than the actual thalweg $S_0$ because of what happens to $S_0(x)$ and $\partial h_0(x)/\partial x$ for topography varying sharply over short distances, as illustrated in Fig. 3. As discussed in

detail in §2 below, non-smoothness in the non-conservative source terms – e.g., RHS of eq. (2) – for a boundary-initial-value problem in partial differential equations presents numerical challenges. If we can discard our (wrong) intuition that the $S_0$ form must somehow "better" represent the variable topography – i.e., recognizing the algebraic equivalence of eq. (5) with eqs. (1) and (2) – it follows that splitting of the Piezometric head to include a body force that is everywhere aligned with a variable $S_0$ is merely creating an unnecessary complexity. Indeed, it is likely this is a reasons why the popular HEC-RAS river hydraulics

model uses an $\eta$ gradient formulation rather than an $S_0$ approach (Brunner, 2016a).

The use of $S_R$ rather than $S_0$ in the governing equations can perhaps be better understood if we think of the slope in eq. (4) as representing simply a portion of the overall Piezometric pressure gradient that can be extracted from $\partial\eta/\partial x$ and treated as a body force that varies gradually along the channel. Hence, in Fig. 3 we are not interested in separating out the details of the sharply-varying slope changes of the local topography, but instead prefer a body force term that aligns with the mean slope

over some larger spatial scale, e.g. as in Fig. 2.

In this paper we examine the effect of variable cross-section geometry on numerical solutions of the SVE and propose a new "Reference Slope" approach that can be inferred from the above arguments. The use of a Reference Slope as a body force





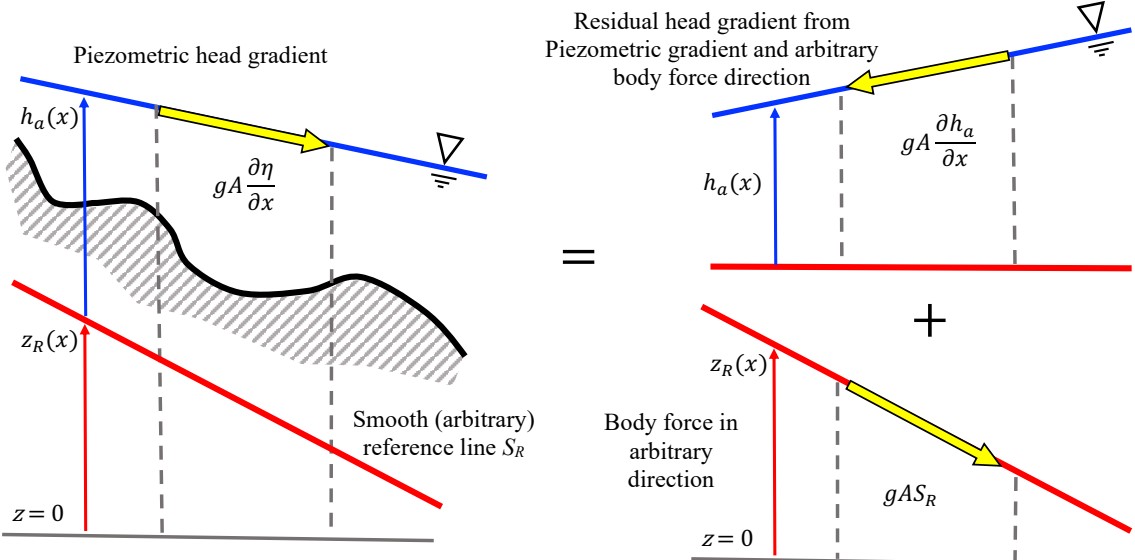

**Figure 2.** Comparison of Piezometric forcing terms for varying topography. On the left is eq. (1) with the arbitrary $S_R$ line provided for reference. The right shows the equivalent split form of forcing with eq. (5) using the identity of eq. (4). The physical bottom topography (shown only on the LHS for clarity) only plays a role through the cross-sectional area ($A$) which feeds back into the solution of variability in $h_a$ in both forms of the equation.

direction to split the Piezometric head gradient term ensures: (i) the slope used in the discrete source term is smooth, (ii) the variable geometry is correctly retained, (iii) the fundamental governing equations are preserved, and (iv) an SVE numerical
algorithm developed using $S_0$ is essentially unchanged. We further demonstrate that bottom-slope discontinuities are a cause of problems in finite-difference forms of 1D Saint-Venant equations with subcritical flow. Of course, this idea will not be a surprise to many modelers who routinely remove troublesome cross-sections or smooth their topography; however the concept does not appear to have been conclusively demonstrated in the literature. More importantly, we show that the problem is inherent in the traditional formulation of the governing equations using the thalweg bottom slope, $S_0$, which is usually computed as
the slope between the lowest points in two adjacent river cross sections. Problems associated with slope discontinuities can be fixed within the governing equations by careful selection of smoothly-changing reference elevations along the channel, $z_R(x)$, which result in smooth reference slopes, $S_R(x)$, and redefinition of the thalweg depth ($h_0$) as a depth associated ($h_a$) with the reference elevation. Note that $h_a$ is not necessarily any characteristic depth of the flow and is only indirectly related to the hydrostatic pressure.
The approach proposed herein can be implemented within any Saint-Venant model as it is entirely independent of the solution algorithm; however, implementation does require re-writing code for the relationships between cross-section area,





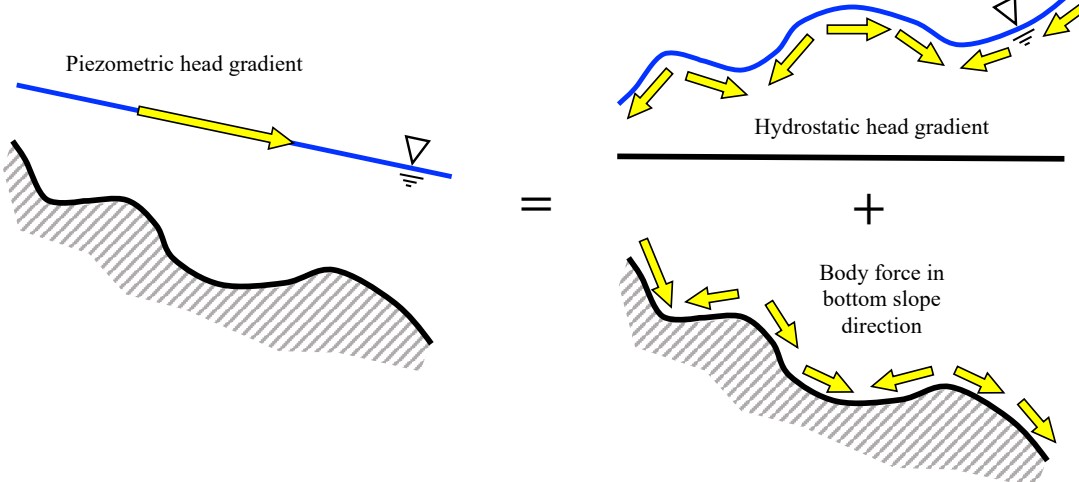

**Figure 3.** Comparison of Piezometric forcing terms with sharply-varying topography for eqs. (1) and (2). The left shows Piezometric head gradient for eq. (1). The right shows the equivalent split form of forcing from eq. (2) using the identity of eq. (3). The variability of the bottom on the LHS will be reflected in the cross-sectional area, $A(x)$, that feeds back into solution of $\eta(x)$. In contrast, the RHS attempts to directly represent topographic variability as both a driving term in $S_0(x)$ and a solution response term $h_0(x)$

wetted perimeter and the "depth" variable of the solution. Note that the new approach includes a re-definition of the depth variable (traditionally the maximum depth, $h_0$) as the depth "associated" ($h_a$) with the reference elevation $z_R$.

## 2 Background

One-dimensional (1D) hydrodynamic models using the Saint-Venant equations (SVE) are widely employed for studying both natural streams and man-made channels (e.g., Martinez-Aranda et al., 2019; Sanders, 2001). It is widely recognized that numerical solutions of the SVE are prone to spurious oscillations in the free-surface elevation unless particular care is taken in the numerical formulation and/or the problem definition (e.g., Nujic, 1995; Tseng, 2004). Numerous techniques and special numerical schemes have been previously designed to overcome unwanted numerical oscillations caused by discontinuous

geometries and boundary conditions (e.g., Zhou et al., 2001; Liang and Marche, 2009). These approaches typically rely on the concept of a "well-balanced" discrete form Greenberg and LeRoux (1996) as discussed in a comprehensive review by Kesserwani (2013) and further elaborated by Hodges (2019). Unfortunately, many water resources models do not use well-balanced schemes, and those that do are often computationally intensive and therefore impractical for simulating regional-to-continental scale river networks or stormwater systems for megacities. When a large-scale open-channel model develops

oscillations and/or instabilities, practitioners may resort to the traditional approach of removing cross-sections or smoothing



bathymetry to mitigate oscillatory or unstable solution behavior (Tayfur et al., 1993). Such *ad hoc* efforts can be effective as they address a major cause of such oscillations and instabilities (discontinuous topography), but they inherently reduce the fidelity of the simulation.

Oscillations and instabilities can be induced in any numerical solution of a boundary-initial-value problem by the inclusion of non-smooth source terms; i.e., if we consider an advection equation of the form

$$\frac{\partial Q}{\partial t} + \frac{\partial}{\partial x}\left(\frac{Q^2}{A}\right) = \sigma \tag{6}$$

where $\sigma$ is a non-homogeneous source term, a fundamental theorem for differential equations provides that a unique solution *cannot* be guaranteed to exist unless the source term is Lipschitz continuous (e.g., Iserles, 1996). Thus, when the thalweg bottom slope ($S_0$) appears as a source term in the SVE it should be *a priori* Lipschitz smooth or oscillations and instabilities

should be expected. For natural systems $S_0(x)$ is typically defined using the maximum channel depth at each surveyed cross-section, which is rarely a smooth function – unless the distance between cross-sections is large compared to bottom elevation variability. Where cross-sections are surveyed at short distances $S_0$ will tend to have significant variability. It follows that the use of $S_0$ has the undesirable property that smaller $\Delta x$ (i.e., resolving a river with more detailed survey data) will increase the non-smoothness in this source term of the momentum equation, resulting in a model that is unlikely to converge under a

grid refinement test. It is not surprising that $S_0$ smoothness, where it occurs in a model of a natural river channel, is typically the result of relatively long separations ($\Delta x$) between cross-section surveys that ensures that the discrete $d^2 z_0 / dx^2$ is small. Thus, removing cross-sections can be an effective mitigation technique because it increases $\Delta x$ and effectively smooths $S_0$. In general, models discretized with higher-resolution river surveys (smaller $\Delta x$) will have greater non-smoothness in $S_0$ and develop more oscillation and instability issues. In essence, our models get worse as our boundary condition data gets better.

The problems associated with $S_0$ can be understood by considering the identity in eq. (3) for a channel with subcritical flow where the free surface curvature is expected to be negligible, i.e., $\partial^2 \eta / \partial x^2 \sim \epsilon$. Taking the along-channel gradient of eq. (3) implies that

$$\frac{\partial^2 h_0}{\partial x^2} = \frac{dS_0}{dx} - \epsilon \tag{7}$$

Thus, forcing with a non-smooth $dS_0(x)/dx$ will require non-negligible curvature of the response variable $h_0(x)$, whose gra-

dient is also a forcing function of the nonlinear equation. Feedback can easily build and cause successive overshoot/undershoot effects, producing oscillations and non-convergence in a nonlinear solver. In contrast, eq. (4) can be invoked with $dS_R/dx$ guaranteed to be small, which implies that $\partial^2 h_a / \partial x^2$ will also be small *even when $\partial A / \partial x$ is non-smooth.* As a practical matter, any $S_0$ with a discontinuous discrete first derivative (i.e., discontinuities in the second derivative of the thalweg elevation, $d^2 z_0 / dx^2$) will be Lipschitz discontinuous and should not be directly discretized in an SVE solution with eq. (2). Although

approximate numerical solutions of equations with non-smooth $S_0(x)$ can sometimes be attained for models with sufficient damping, such solutions are questionable as they do not have rigorous mathematical foundations.

  Arguably, non-smoothness in $S_0$ can be handled in one of four ways: (i) smoothing the geometry – hence solving for flows that do not match the real system; (ii) applying *ad hoc* smoothing within the flow solution – i.e., adjusting the physics to remove





numerical instabilities; (iii) adjusting the numerical discretization scheme to compensate for non-smoothness – e.g., the well-
balanced concept; or (iv) adjusting the governing equations to ensure that any slope in the source term is smooth *without*
modifying the solution physics, the channel geometry, or the numerical discretization scheme. It should be obvious that eq. (1)
is the extreme example of the last approach – replacing $S_0$ and $\partial h_0/\partial x$ with $\partial \eta/\partial x$ ensures that $S_0$ does not occur in the
governing equations and cannot destabilize the solution. To our knowledge the last approach (as used herein and illustrated
in Fig. 2) has not been previously proposed or analyzed in the literature. Nevertheless, as shown below it provides a simple
method that can be readily adapted into existing hydrodynamic models.

For brevity, we will limit our focus herein to subcritical flows – backwater tends to smooth the effects of slope discontinuities
and thus we expect smooth solutions for flow rate and free-surface elevation despite non-smooth geometry. Nevertheless,
common SVE solvers can exhibit oscillatory, non-convergent behavior even in simple subcritical flows when geometry is not
smooth. In the following it will be obvious that the mathematical theory applies directly to supercritical and transcritical flows
as well, but evaluating model performance under the breadth of possible transcritical conditions (including non-smooth jumps)
necessarily requires more analyses than is practical in a single paper.

## 3 Methods

### 3.1 SPRNT

The Simulation Program for River Networks (SPRNT) code for unsteady SVE river networks is used and modified herein. The
baseline for this code models momentum using eq. (2), which is coupled to solution of continuity

$$\frac{\partial A}{\partial t} + \frac{\partial Q}{\partial x} = q_\ell \tag{8}$$

where $q_\ell$ is a lateral inflow per unit length. Note that significantly non-smooth $q_l(x,t)$ can provide another source of numerical
oscillations and instability (Kuiry et al., 2010). As the main focus of this study is the slope source term in the momentum
equation, the lateral inflows and their effects are neglected by setting $q_\ell = 0$ everywhere.

In the SPRNT momentum equation, the friction slope is represented as

$$S_f = \frac{n^2 P_w^{4/3}}{A^{10/3}} Q^2 \tag{9}$$

where $P_w$ is the wetted perimeter of a cross-section and $n$ is Manning's $n$. Although, there are other methods for treating
frictional losses (e.g., Decoene et al., 2009; Burguete et al., 2007), the Chezy-Manning formulation remains popular due to its
simplicity.

The baseline model uses the thalweg elevation ($z_0$), the thalweg depth ($h_0$), and the thalweg bottom slope ($S_0$) as

$$h_0 \equiv \eta - z_0 \tag{10}$$

$$S_0 \equiv -\frac{\partial z_0}{\partial x} \tag{11}$$





SPRNT is an open-source, 1-D hydrodynamic solver using the fully-implicit Preissmann numerical scheme (Preissmann, 1961) with Newton-Raphson iteration and computational acceleration techniques developed from Very Large Scale Integration

(VLSI) semiconductor design. Details on the baseline SPRNT model and its application to large river networks are provided in Liu and Hodges (2014) and Yu et al. (2017).

### 3.2  Reference Slope (RS) method

We introduce a a new "Reference Slope Method" (RS) through a transformation and redefinition of geometry in the Saint-Venant equations as discussed in §1. In place of the conventional $h_0$ and $z_0$, we define a "reference elevation" ($z_R$) and its

"associated depth" ($h_a$) as shown in Fig. 4. These provide a relationship with the free surface elevation ($\eta$) defined as

$$h_a \equiv \eta - z_R. \tag{12}$$

Note that $z_R$ is arbitrary, so $h_a$ may be either greater than or less than the thalweg depth $h_0$ at a given location. As shown in Fig. 4, it is convenient to define the "reference height" ($h_R$) in relation to $z_R$ and the true bottom elevation, $z_0$, by

$$h_R \equiv z_0 - z_R \tag{13}$$

Thus, the conventional $h_0$ and $z_0$ are recovered with

$$h_0 = h_a - h_R \tag{14}$$
$$z_0 = z_R + h_R \tag{15}$$

We define the reference slope ($S_R$) as the downstream slope of $z_R$:

$$S_R \equiv -\frac{\partial z_R}{\partial x} \tag{16}$$

Using eqs. (12) and (16) in eq. (2) provides eq. (5), which is more conveniently written as

$$\frac{\partial Q}{\partial t} + \frac{\partial}{\partial x}\left(\frac{Q^2}{A}\right) = -gA\frac{\partial h_a}{\partial x} + gA(S_R - S_f) \tag{17}$$

The above is identical to eq. (2) with the simple substitution of $h_a$ and $S_R$ for $h_0$ and $S_0$. In this formulation, the definition of $z_R(x)$ is arbitrary, so we can *a priori* require its selection such that $S_R(x)$ is smooth. A trivial choice that is guaranteed smooth is $z_R(x) = $ constant, which returns $S_R(x) = 0$ and the Saint-Venant equations in the form of eq. (1). For the present purposes

we are interested in non-trivial definitions of $z_R(x)$ that are close to $S_0(x)$ but are guaranteed smooth. If $S_R(x)$ is smooth then the source term of the equation can be guaranteed smooth as long as $S_f(x)$ is smooth – which is typically true as long as the solution $Q(x)$ is smooth. Note that in extreme cases of geometric discontinuity the values of $n$, $P_w$ and $A$ in eq. (9) can cause a non-Lipschitz source term; however, most solution methods are relatively robust to such discontinuities as they are in the coefficient of the solution variable rather than an additive source term.





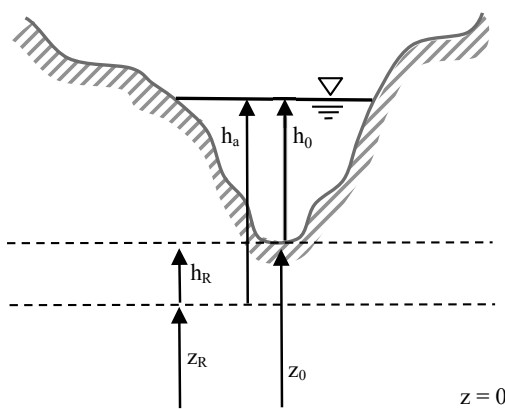

**Figure 4.** Relationships of $h_a$, $h_R$, $z_R$ and $z_0$ for an arbitrary cross-section. Note that $z_R > z_0$ is also allowed, which results in $h_a < h_0$ and a negative value for $h_R$. Furthermore, if $z_R > z_0 + h_0$ then $h_a$ is negative to retain algebraic consistency. Modified from Liu (2014), used by permission.

A critical change required by the introduction of $h_a$ is that the conventional geometric auxiliary relationships of $A = f(h_0)$ and $P_w = f(h_0)$ must be transformed in $A = f(h_a)$ and $P_w = f(h_a)$. That is, once we change the depth in our equation from $h_0$ to $h_a$ we must re-index the geometry. In general, for known functions $A(h_0)$ and $P_w(h_0)$ this is a trivial transformation as

$$A = A(h_a - h_R) \tag{18}$$

$$P_w = P_w(h_a - h_R) \tag{19}$$

However, the implementation in an existing code is not necessarily as simple as the above equations suggest. For example, in Fig. 4, the condition $A = 0$ occurs where $h_a = h_R$, i.e., a non-zero value as compared to $h_0 = 0$ with conventional geometry. Unfortunately, model developers typically have *ad hoc* wetting/drying treatments that are introduced as $h_0 \to 0$ or for $h_0 < 0$. Such treatments need to be modified to deploy as $h_a \to h_R$; which introduces the added complication that $h_R$ is negative where $z_R > z_0$. Note that the new geometry does not require altering the wetting/drying algorithms itself or, for that matter,

any other solution algorithm – only the actual geometry definitions require alteration. These relatively straightforward changes can be contrasted with the effort needed to provide a well-balanced numerical discretization scheme for the conventional $S_0$ representation of geometry (e.g., Kesserwani, 2013)

        The modification of the SPRNT code to implement the above RS method will be known as SPRNT-RS. The SPRNT and SPRNT-RS source codes are available in an open-source repository (Liu, 2014). Note that solution algorithm for SPRNT-RS

is identical to that of SPRNT; the only code changes are for the new geometry definitions for $h_a$ and $S_R$ that replace $h_0$ and $S_0$ in the original algorithm. This simple geometry replacement strategy is effective because eq. (17) is identical to eq. (2) except for the change in nomenclature to $h_a$ and $S_R$.





### 3.3 Generating a smooth $S_R(x)$

The $z_R(x)$ and hence $S_R(x)$ are arbitrary choices in the RS method, but should be generated for smoothness of $S_R(x)$ along
the reach. In synthetic reach test cases and analytical test cases (described below), the channels are *a priori* either specified with
a uniform $S_R$ or produced by different order of splines. For our urban creek test case the $z_R$ is generated with an approximating
cubic B-spline (de Boor, 2001) based on thalweg, $z_0(x)$, elevations. There are a variety of possible ways to generate smooth
$S_R(x)$, but applying approximating cubic B-splines to the $z_0(x)$ is convenient because the slope is guaranteed to be locally
smooth as long as the knot spacing in the B-spline is everywhere larger than the spacing between cross sections. It should be
emphasized that an exact spline fitting of all the thalweg data (i.e., knots at all the cross-sections) will be smooth at scales finer
than the cross-section spacing but non-smooth at the model's discretization scale. That is, exact cubic spline fitting of $z_0(x)$
does *not* reduce discontinuities at the discretization scale – only an approximate fitting associated with coarser scales than the
cross-sectional spacing will be effective.

### 3.4 HEC-RAS for model validation

The baseline SPRNT has been previously shown to have excellent agreement with the Hydrologic Engineering Center River
Analysis System (version 5.0.7) – known as HEC-RAS. Liu and Hodges (2014) showed SPRNT simulations agreed with HEC-
RAS with $\leq 3\%$ difference in water depth solution when using both prismatic cross-sections and nonuniform channels. Thus,
HEC-RAS provides a reasonable model for testing and validation of SPRINT-RS. We would have preferred to use a single
model with and without the RS method for such model-model comparisons; however, HEC-RAS is a closed-source proprietary
model so we could not directly implement and test the RS method in that code. Conversely, as expected by the discussions in
§2, the baseline SPRNT model is oscillatory and non-convergent on the highly-discontinuous geometry of our test cases due to
its use of the $S_0$ approach, so it cannot be directly used for before/after comparisons of RS. Thus, simulations using SPRNT-RS
are compared to HEC-RAS simulations for validation and insight.

HEC-RAS provides a convenient validation model for three reasons. Firstly, it is a widely-accepted engineering model for
river-reach simulations, (e.g., Wang et al., 2012; Giustarini et al., 2011; Aggett and Wilson, 2009). Secondly, it has been used
as a validation model in numerous prior studies (e.g., Gichamo et al., 2012; Mejia and Reed, 2011; Horritt and Bates, 2002).
Finally, unsteady HEC-RAS employs $\partial \eta / \partial x$ as the piezometric gradient rather than using $\partial h_0 / \partial x$ and $S_0$, which is one of the
reasons it is relatively robust for non-smooth geometry such as used herein.

The performance of the RS method is demonstrated below through: (i) comparison to six analytical test cases from Mac-
Donald et al. (1995) with Lipschitz-continuous geometry, various prismatic shapes, and different formulations of $S_R$; (ii) seven
synthetic test cases using Lipschitz-discontinuous geometry; and (iii) an urban creek with complex cross-section geometry de-
rived from physical surveys that include discontinuities an order of magnitude greater than those in the synthetic test cases.





## 3.5 Test cases – analytical solutions

Analytical solutions of six test cases with different channel shapes and bed slope formations from MacDonald et al. (1995) are
used to show that SPRNT-RS reproduces the correct water surface elevation regardless of the selection of $S_R$. These test cases
are representative of the more comprehensive analysis provided in Yu et al. (2019). The configuration details for each case are
provided in Table 1, where we adopt the nomenclature of MacDonald et al. (1995) for ease of comparison. The selected test
cases have Lipschitz-smooth geometric features that are represented in RS tests using both uniform and splined reference beds,
as shown in Fig. 5. To illustrate the adaptability of the RS method, the UR3 and UT2 cases use splines that produce $z_R$ very
close to (but not identical to) the actual bed, whereas the other cases use uniform $S_R$ or splines with greater differences. The
uniform $S_R$ in cases UR1, UT1, and VR1 are set to the average slope in each domain. With reference to Fig. 4, the differences
between the channel bottom ($z_0$) and the reference bottom ($z_R$) shown in Fig. 5 imply channel bottom offsets ($h_R$) of varying
complexity for the RS method, as shown in Fig. 6. The VR1 and VR2 cases also provide smooth changes in the channel
width, which are shown in Fig. 7.The node spacing for all these tests is a uniform $\Delta x = 10$ m. The boundary conditions follow
MacDonald et al. (1995).

| Case name | Cross section shape type | Cross section shape detail | $n$ | $Q$ | $S_R$ |
|-----------|--------------------------|----------------------------|-----|-----|-------|
| UR1 | Uniform rectangular | $W_B = 10$ m | | | Constant $z_R$ |
| UR3 | Uniform rectangular | | | | 1st order spline |
| UT1 | Uniform trapezoidal | $W_B = 9$ m, $S_{SW} = 2$ | 0.03 | 20 m³/s | Constant $z_R$ |
| UT2 | Uniform trapezoidal | $W_B = 10$ m, $S_{SW} = 2$ | | | 2nd order spline |
| VR1 | Varying rectangular | Varying $W_B$ | | | Constant $z_R$ |
| VR2 | Varying rectangular | | | | 3rd order spline |

**Table 1.** Configuration and geometric data for analytical test cases derived from MacDonald et al. (1995). $W_B$ and $S_{SW}$ represent bottom
width and sidewall slope, respectively.





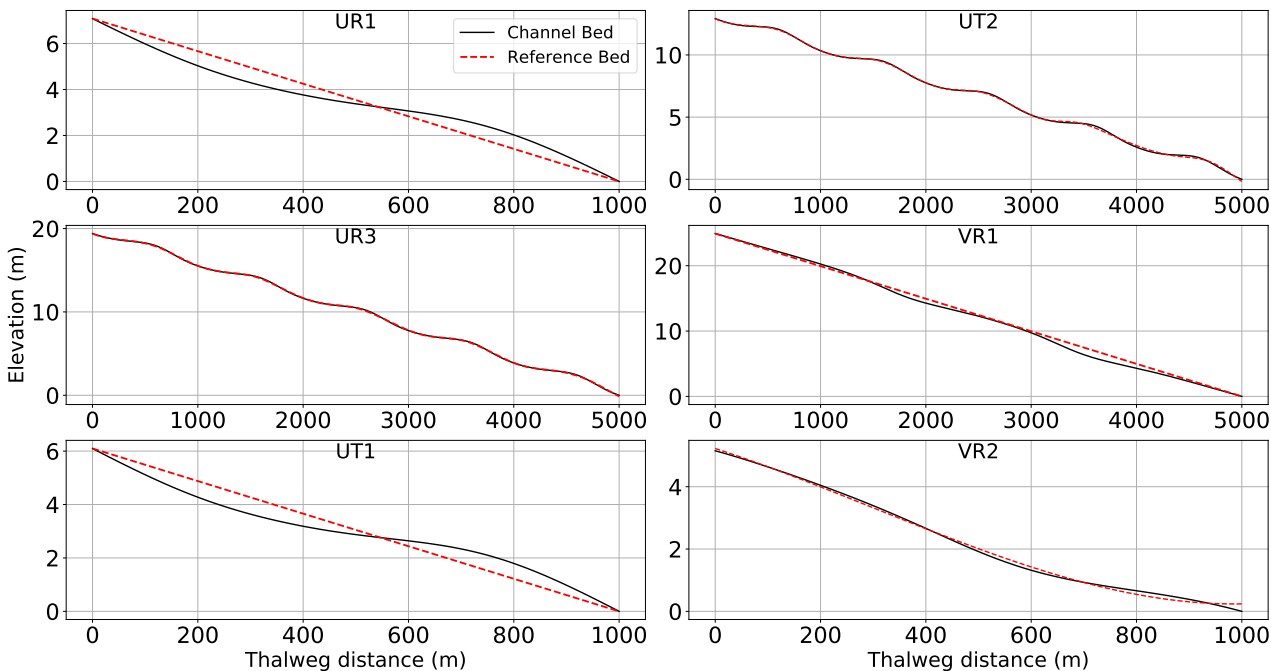

**Figure 5.** Channel bed elevation ($z_0$) and reference bed elevation ($z_R$) for six test cases from MacDonald et al. (1995).

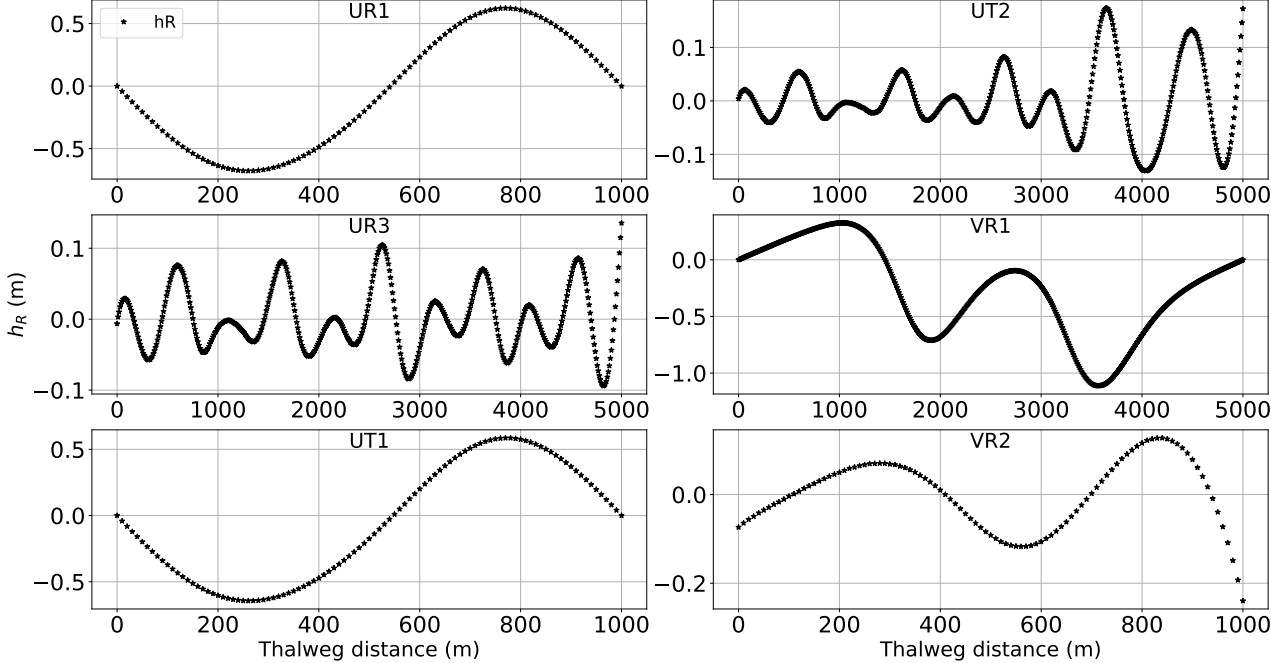

**Figure 6.** Reference bed offset, $h_R$, for $z_R$ and $z_0$ of the test cases in Fig. 5.





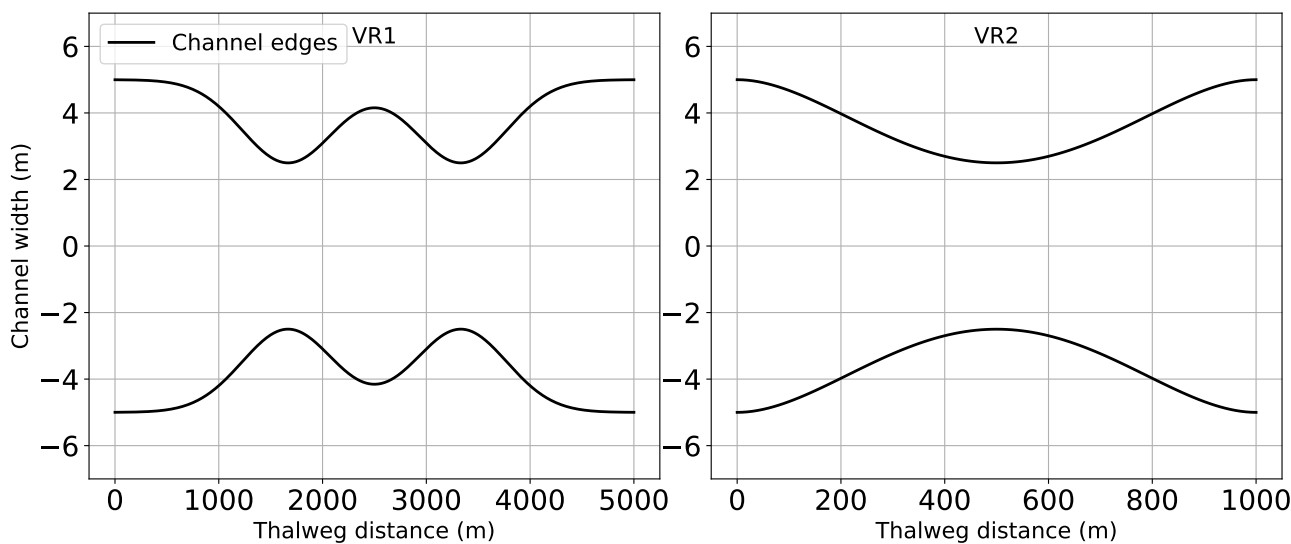

**Figure 7.** Plan view of channel edges for cases VR1 (left) and VR2 (right) of Fig. 5.



## 3.6 Test cases – synthetic channel reach

The analytical test cases, above, are designed to show that the RS method does not introduce approximations that affect the smooth solution. However, the true power of the RS method is in solution of non-smooth bathymetry where most models using $h_0$ and $S_0$ have difficulty converging. To illustrate this aspect we use a simple river reach with randomly-perturbed

(discontinuous) bathymetry at various scales. As there are no analytical solutions for these tests we use the HEC-RAS model for comparison. The simulations use time-invariant boundary conditions with geometry defined by trapezoidal cross-sections of uniform side-slope, as detailed in Table 2. The channel bed offset ($h_R$) and thalweg slope ($S_0$) are illustrated in Fig. 8. The flow boundary conditions provide for a mild slope with a water surface profile that can be classified as an M1 gradually-varying flow.

Case 1 is the baseline smooth channel with a uniform slope over the entire reach length. Cases 2 through 5 have synthetic geometry developed by random perturbations of the bottom elevation of the baseline reach. Cases A and B have the identical smooth geometry to Case 1, but use different reference slopes for RS tests. The synthetic channel test reach is 1.58 km in length discretized into 80 uniform computational nodes with 79 channel segments (20 m per segment). The trapezoidal cross-sections each have a 10.0 m bottom width and 63.4 degree sidewall slopes. Bottom roughness is fixed by a Manning's $n$ of 0.04 for all

segments.

| Case | $\alpha$ | $h_R$ | $S_0$ | $S_R$ |
|---|---|---|---|---|
| Case 1 (baseline) | – | – | 0.008 | 0.008 |
| Case 2 | 0.01 | [ $-0.0018$, 0.0012] | [0.0079, 0.00802] | |
| Case 3 | 0.1 | [ $-0.0186$, 0.0126] | [0.0077, 0.0082] | 0.008 |
| Case 4 | 0.5 | [ $-0.0914$, 0.0632] | [0.0067, 0.0092] | |
| Case 5 | 1 | [ $-0.1827$, 0.1264] | [0.0056, 0.0105] | |
| Case A | 0 | [ $-3.2$, 3.12 ] | 0.008 | 0.004 |
| Case B | 0 | [ $-1.56$, 1.6] | 0.008 | 0.010 |

**Table 2.** Configuration of synthetic channel reach test cases: $\alpha$ is used in eq. (20) for random perturbation of the baseline Case 1 geometry; $h_R$ is the bed offset based on Fig. 4 with brackets indicating upper and lower limits of randomized geometry values over the non-uniform test reach; $S_0$ is the range of the thalweg slope for $z_0$ from eq. (20). $S_R$ is the selected uniform reference slope.

To develop the random perturbations of the bottom in the synthetic test cases, we being with the Case 1 (baseline) using a uniform $S_0^{[1]} = 0.008$ over the entire reach length. Here the superscript in square brackets indicates the case identifier. The set of bottom elevations for Case 1 are $z_0^{[1]}(x)$, which are smooth and linearly decreasing over the reach length. Cases $2 - 5$ are similar channels with perturbed bottom elevations set by

$$z_0^{[c]}(x) = z_0^{[1]}(x) + \alpha^{[c]} H(x) \quad : \quad c \in \{2, 3, 4, 5\} \tag{20}$$



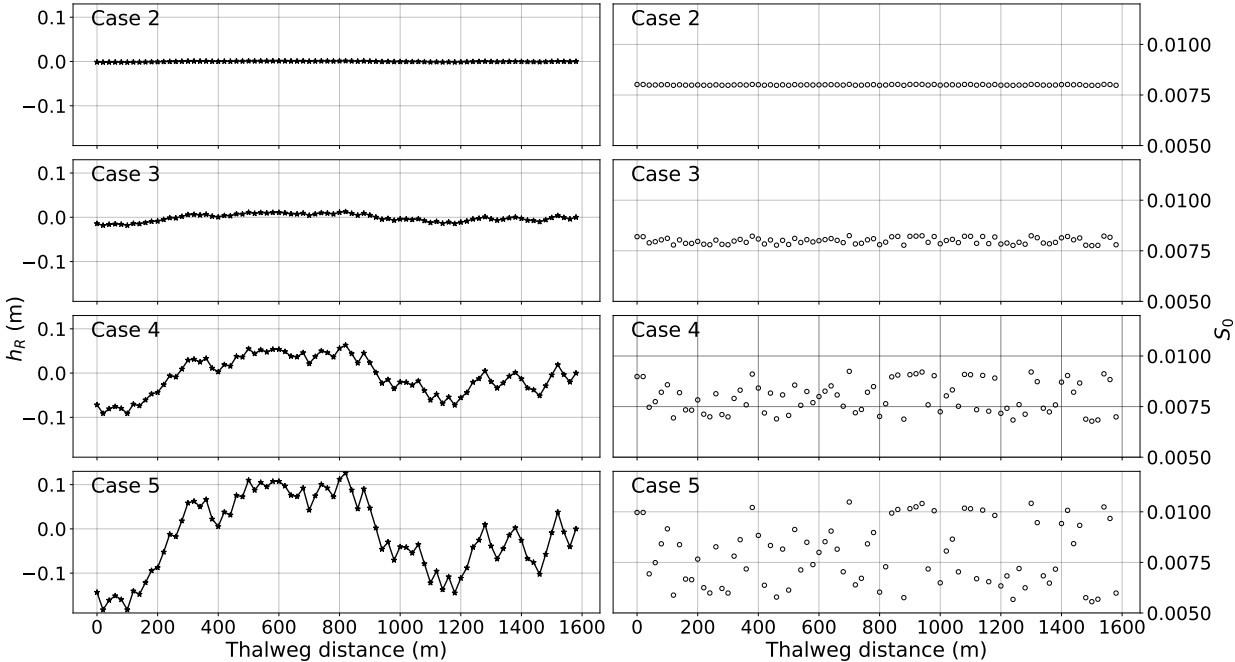

**Figure 8.** Channel bottom offset ($h_R$) and physical thalweg slope ($S_0$) for synthetic test Cases $2-5$ that are random perturbations of the baseline smooth $S_0 = 0.008$ of Case 1.

where $H(x)$ is a set of random-generated numbers within the range of $-0.126 \leq H(x) \leq 0.183$. The upper/lower limits of the $H(x)$ were selected to prevent the occurrence of a locally-adverse slope – such conditions can be handled by SPRNT-RS but can cause convergence problems for some models. The $\alpha^{[c]}$ is a magnitude to generate a range of bottom displacements with $\alpha^{[c]} \in \{0.01, 0.1, 0.5, 1.0\}$ for $c \in \{2, 3, 4, 5\}$ respectively. Cases $2-5$ set the reference bottom elevations exactly equal to the baseline Case 1 physical bottom elevations, i.e.

$$z_R^{[c]}(x) = z_0^{[1]}(x) \quad : \quad c \in \{2, 3, 4, 5\} \tag{21}$$

Thus, the SPRNT-RS simulations for Cases $2-5$ use uniform $S_R$ over the reach such that the bed offset ($h_R$) represents the physical geometric perturbations. Noting from eq. (13) that $h_R$ is the difference between the physical bottom ($z_0^{[c]}$) and the reference bottom ($z_R^{[c]}$), substituting the above relationships provides

$$h_R^{[c]}(x) = \alpha^{[c]} H(x) \tag{22}$$

For synthetic test Cases A and B in Table 2, the actual channel bottom slopes are set to uniform values equivalent to Case 1; that is $S_0^{[A]} = S_0^{[B]} = S_0^{[1]} = 0.008$, with identical thalweg elevations of $z_0^{[1]}(x)$. However, the reference slopes for these cases are set to smaller and greater uniform values: $S_R^{[A]} = 0.004$ and $S_R^{[B]} = 0.010$. These two cases demonstrate the RS method generates the same numerical solution as baseline Case 1 (solved at $S_0$) when $S_R$ is set to an arbitrary value.





Forcing for all seven test cases is a constant inflow boundary of 283 $m^3s^{-1}$ applied at the furthest upstream node. The downstream boundary condition is 5.0 m of depth, which is subcritical flow based on a normal depth of 4.95 m for $S_0 = 0.008$ and the inflow rate. Because a subcritical boundary condition allows upstream wave reflections, the downstream boundary was enforced at the end of a 180 m (9 node) buffer domain, which was adequate for reducing upstream wave propagation in unsteady flow solutions. Simulation results are reported after the models have reached a steady state and all oscillations

associated with the initial conditions have dissipated. Solutions for the buffer segment are not included in the analyses below.

### 3.7    Test case – Waller Creek study site

The main stem of Waller Creek in Austin, Texas (USA) is used to examine the performance of the RS method for more complex conditions. The main stem of creek drains an urban watershed of 14.3 $km^2$ with total length of 10.7 km for the area illustrated in Figure 9. Bathymetric survey data are available courtesy of the City of Austin (Figure 10). The bathymetric data set includes

327 surveyed cross-sections with spacing intervals ranging from 2.5 m up to 178 m (mean of 33.5 m). The Manning's $n$ of the channel (based on City of Austin computations) varies from 0.02 to 0.06 throughout the system. The SPRNT model, in both its original and RS form, has numerical stability issues associated with close cross-sectional spacing. Arguably, these issues are related to sharp changes in $A$ that lead to non-smooth source terms despite the RS method – however, this issue requires further investigation. Similar numerical instability behavior can also be found in the HEC-RAS unsteady model and

also causes divergent solutions. For the present work, we discarded 36 cross-sections (11% of the data set) that were closer than 10 m and merged these short reach lengths with the adjacent sections. An additional three cross-sections were discarded and some channel roughness values were modified as they cause numerical instability in the HEC-RAS unsteady simulation (see Appendix for details). The resulting data set is 288 cross-sections with spacing ranging from 10.1 m to 184.9 m. The mean cross-section spacing is 37.2 m with a total reach length of 10.7 km. To limit our focus to subcritical flow, our analyses consider

only the upper 8.3 km of the main reach (210 out of 288 cross-sections), which eliminates a series of step-pool transcritical elements in the downstream channel where the HEC-RAS solution is strongly influenced by the *ad hoc* LPI algorithm (Fread et al., 1996; Brunner, 2016b). The smoothing introduced by LPI makes it difficult to draw conclusions from a comparison between SPRNT-RS and HEC-RAS across transcritical locations.

    A time-invariant upstream inflow boundary condition is set to 25 $m^3s^{-1}$ at the headwater cross-section. To minimize the

influence of subcritical reflections from the upstream inflow boundary, the first 10 computational nodes at the upstream are not included in the results analysis. Lateral inflows are set to zero for all test cases. A 300 m buffer section is added downstream of the test domain to reduce the influence of reflections from the downstream boundary condition. This buffer section uses the same cross section as the final downstream section of the data set with bed slope ($S_0$) of 0.0033 and Manning's $n = 0.04$. The buffer section has a normal depth of 0.76 m at the 25 $m^3s^{-1}$ inflow rate. The downstream boundary condition at the end of

the buffer section is 0.7 m depth, which is subcritical and implies an M2 gradually-varying drawdown in the vicinity of the outflow. These geometry and boundary conditions are identically applied to both SPRNT-RS and HEC-RAS models.

    The thalweg elevation, $z_0(x)$, and the reference elevation, $z_R(x)$, of the RS method (as determined by the approximate spline fit described above) for Waller Creek are shown in Fig. 11(a). The $z_0$ and $z_R$ are visually similar with the former being



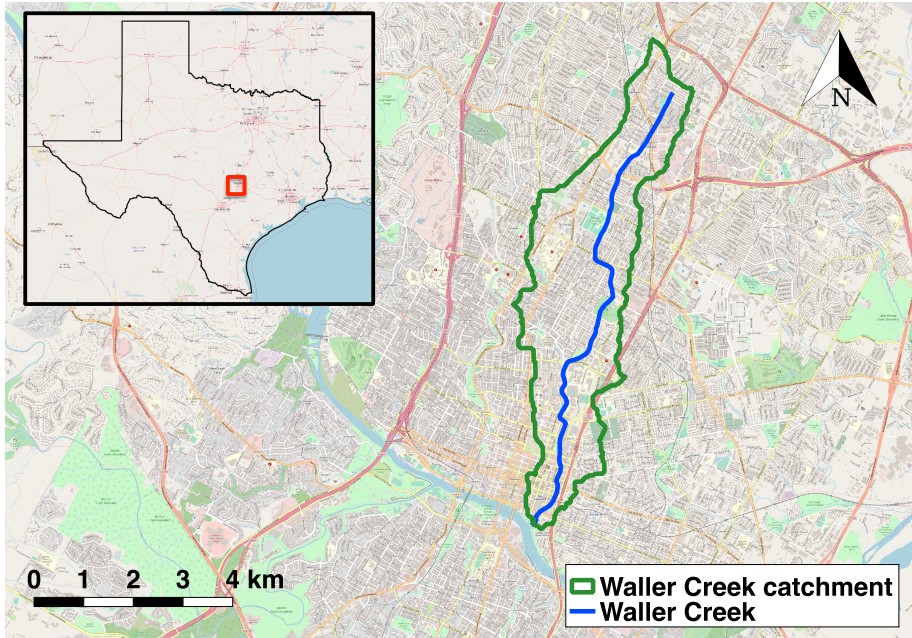

**Figure 9.** Main stem of Waller Creek and catchment in Austin (Texas, USA). Basemap copyright by Open Geospatial Consortium (OGC) Web Mapping Service (WMS).

somewhat more noisy. The elevation data sets provide similar overall reach slopes (uppermost cross-section to lowermost
cross-section) of 0.0074 and 0.0077 respectively. Note that the approximate B-spline technique for generating $z_R(x)$ does not force the overall reach slope to be identical. Because the $z_R(x)$ are mathematically arbitrary there is no need to force an exact match. Although $z_0(x)$ and $z_R(x)$ are similar in Figure 11(a), the $S_0(x)$ from the raw data are discontinuous and vary over a wide range (up to $4\times$ the reach slope), as illustrated in Figure 11(b). Note that $S_0(x)$ also includes negative slopes (i.e., adverse gradient sections), which can cause convergence problems for some numerical solvers. In contrast, as shown in
Figure 11(c), the approximate cubic B-spline used to generate $S_R(x)$ from $z_R(x)$ provides a reference slope that is everywhere smooth, positive and remains close to the overall reach slope of 0.008. The slope range and model nomenclature for the Waller Creek test cases are provided in Table 3. The Lipschitz smoothness of $S_0$ versus $S_R$ can be better understood by evaluating the gradient of the slope, i.e., the 2nd derivative of $z_0$ and $z_R$, as shown in Fig. 11(d). The $S_0$ formulation clearly lacks smoothness in the higher derivative.





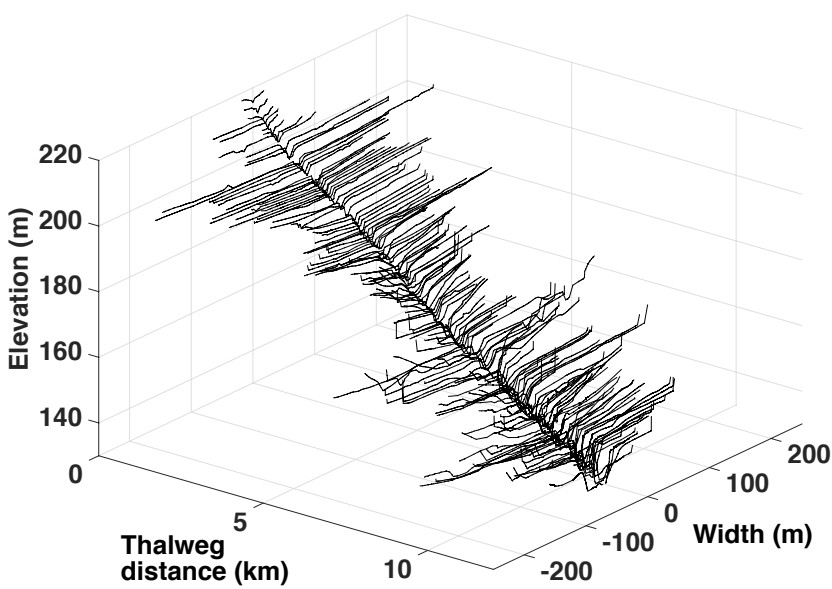

**Figure 10.** Surveyed cross-sections of Waller Creek (Texas). Only 140 out of 327 cross-sections are shown for clarity. Elevations are relative to mean sea level. Data courtesy of City of Austin.

| Case | Slope formulation | Slope range | Model usage |
|------|-------------------|-------------|-------------|
| $WC_{RS}$ | RS method | $0.0033 < S_R < 0.0147$ | SPRNT |
| $WC_{HEC-S}$ | Conventional | $-0.0328 < S_0 < 0.0393$ | HEC-RAS(steady) |
| $WC_{HEC-U}$ | Conventional | $-0.0328 < S_0 < 0.0393$ | HEC-RAS(unsteady) |

**Table 3.** Data for model setup of Waller Creek test cases.

## 3.8 Analysis methods

To evaluate the performance of the RS method relative to conventional formulations, four depth-based indicators are employed, as described below. For these definitions the control (superscript [C]) is the MacDonald et al. (1995) solution for the analytical test case and HEC-RAS results for the synthetic channel and Waller Creek test cases. Note that the synthetic tests use unsteady HEC-RAS whereas the Waller Creek study uses comparisons to both steady and unsteady versions of the model. The test case (superscript [T]) is always the SPRNT-RS simulation. These measures can be considered error metrics for the comparison to the analytical solutions and difference metrics for model-model comparisons.



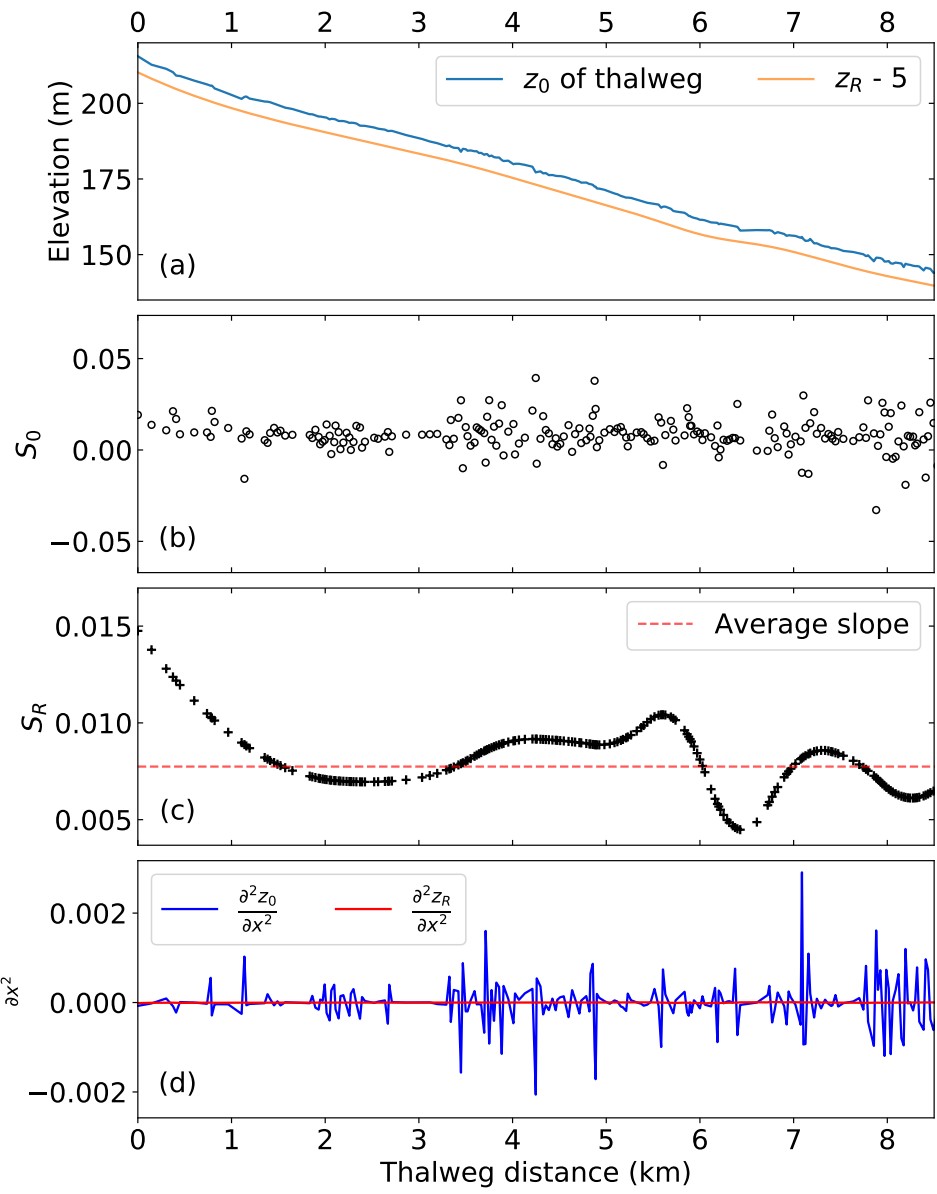

**Figure 11.** Waller Creek bottom elevation and slope: (a) $z$ elevation, with 5 m subtracted from $z_R$ for clarity; (b) $S_0$ thalweg slope between cross-sections; (c) $S_R$ smoothed bottom slope. (d) 2nd derivative of $z_0$ and $z_R$. Note the $y$-axis scaling in (c) has reduced limits compared to (b) to better show the smoothness achieved by the spline fit.

**(1) *Normalized difference* ($\rho$).** A non-dimensional index to describe the local difference in depth can be defined as:

$$\rho^{[T:C]}(x) = \frac{h_0^{[C]}(x) - h_0^{[T]}(x)}{h_0^{[C]}(x)} \tag{23}$$





where $h_0^{[C]}(x)$ and $h_0^{[T]}(x)$ are the local depth solution from the control and test case results after steady-state conditions are
achieved. The normalization scale is the local depth of the control case. Note the denominator is non-zero in the synthetic test
case setup because the flow setup is an M1 gradually-varying flow.

**(2)** *Absolute mean normalized difference* ($\zeta$). The mean of the absolute value of $\rho(x)$ over the domain provides an overall
non-dimensional indicator of the depth error:

$$\zeta^{[T:C]} = \frac{1}{N} \sum_{x=1}^{N} \mid \rho^{[T:C]}(x) \mid \tag{24}$$

where $N$ is the total number of cross-sections. We use the absolute value so that positive errors do not cancel negative errors
and $\zeta$ is a representative scale of the discrepancy between models.

**(3)** *Mean absolute error* **(MAE).** The overall dimensional error is characterized as:

$$\text{MAE} = \frac{1}{N} \sum_{x=1}^{N} \mid h_0^{[C]}(x) - h_0^{[T]}(x) \mid \tag{25}$$

and the non-dimensionalized form of overall error is:

$$\text{MAE (non-dimensional)} = \frac{1}{N} \sum_{x=1}^{N} \mid \frac{h_0^{[C]}(x) - h_0^{[T]}(x)}{h_0^{[C]}(x)} \mid \tag{26}$$

**(4)** *Root-mean-square-error* **(RMSE).** A standard dimensional measure of the squared error is:

$$\text{RMSE} = \sqrt{\frac{1}{N} \sum_{x=1}^{n} (h_0^{[C]}(x) - h_0^{[T]}(x))^2} \tag{27}$$

The non-dimensional form of RMSE is computed by the following equation:

$$\text{RMSE (non-dimensional)} = \sqrt{\frac{1}{N} \sum_{x=1}^{n} (\frac{h_0^{[C]}(x) - h_0^{[T]}(x)}{h_0^{[C]}(x)})^2} \tag{28}$$

Both the MAE and RMSE are also reported in non-dimensional form where the normalization scale is presented.

## 4  Results

### 4.1  Analytical test cases

The water surface elevations for the analytical solutions and SPRNT-RS simulations are shown in Fig. 12. Visually, the analytical and simulated results across all six cases are identical. Error metrics following §3.8 are provided in Table 1. The normalized
differences ($\rho$) are less than 1% and are consistent with absolute errors of $O(10^{-3})$ m, which are negligible compared with
the water depths $\geq 1$ m. The spatial distributions of the normalized error are shown in Fig. 13. By comparing this figure with
Fig. 5 it can be seen that $\rho(x)$ fluctuates with the change of bed slope. Similar behavior can also be found for model results
reported in MacDonald et al. (1995).





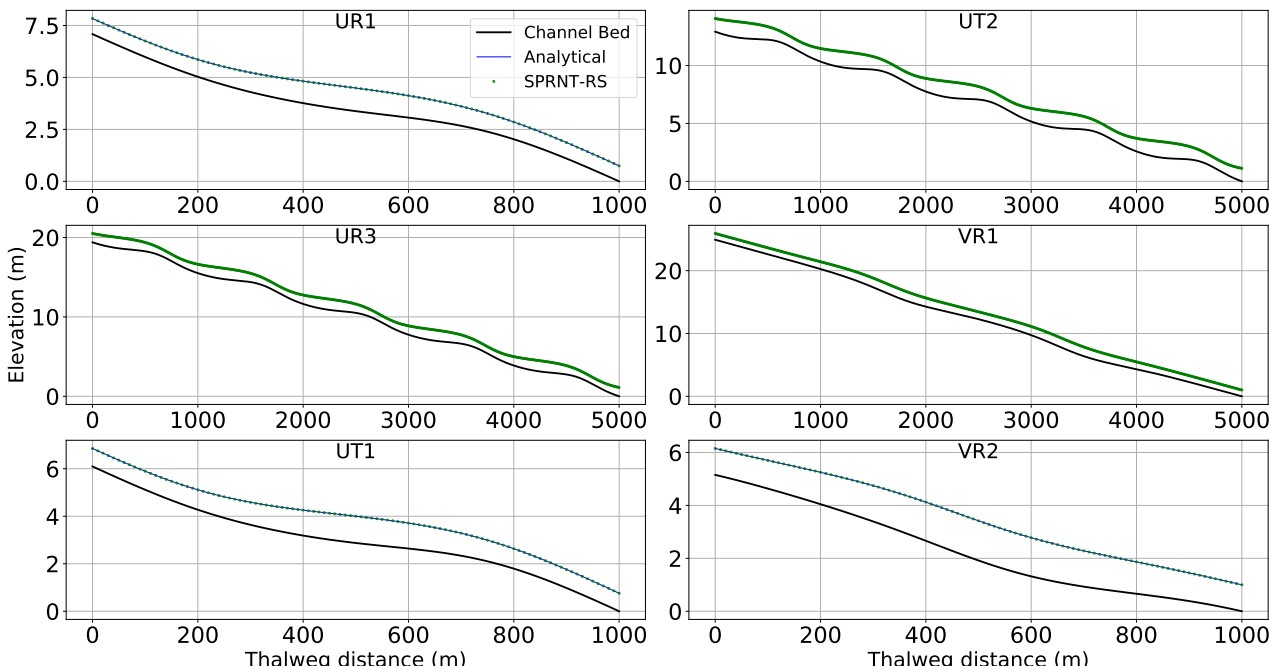

**Figure 12.** Comparison between simulated water surface elevation from SPRNT-RS and analytical solution for analytical test cases of MacDonald et al. (1995).

| Case | min($\rho$) | max($\rho$) | $\zeta$ | MAE (m) | RMSE (m) |
|------|---------|---------|------|---------|----------|
| UR1 | $-0.47\%$ | $0.33\%$ | $0.09\%$ | $0.00104\ (0.032\%)$ | $0.00131\ (0.048\%)$ |
| UR3 | $-0.73\%$ | $0.55\%$ | $0.28\%$ | $0.00385\ (0.044\%)$ | $0.00457\ (0.058\%)$ |
| UT1 | $-0.19\%$ | $0.33\%$ | $0.07\%$ | $0.00085\ (0.027\%)$ | $0.00108\ (0.042\%)$ |
| UT2 | $-0.73\%$ | $0.62\%$ | $0.28\%$ | $0.00388\ (0.064\%)$ | $0.00458\ (0.084\%)$ |
| VR1 | $-0.13\%$ | $0.08\%$ | $0.03\%$ | $0.00055\ (0.006\%)$ | $0.00070\ (0.009\%)$ |
| VR2 | $-0.15\%$ | $0.13\%$ | $0.06\%$ | $0.00090\ (0.026\%)$ | $0.00108\ (0.031\%)$ |

**Table 4.** Difference measures using eqs. (23) – (28) for analytical test cases of MacDonald et al. (1995). Non-dimensionalized MAE and RMSE are shown in parentheses.



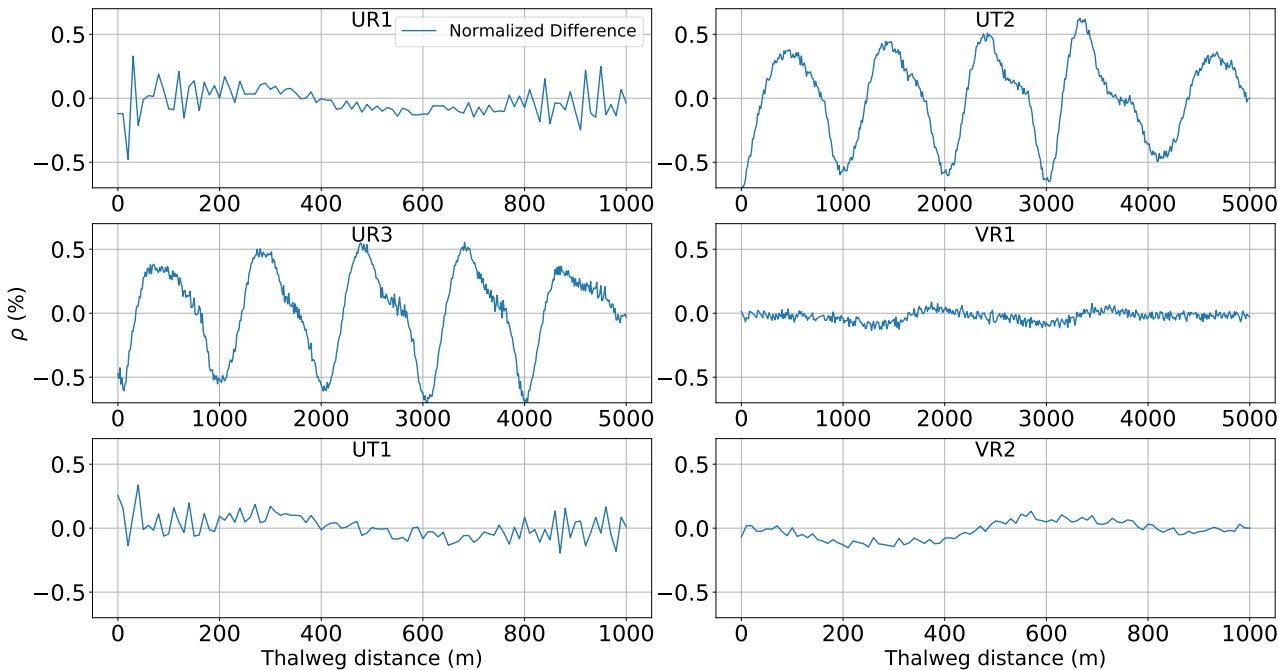

**Figure 13.** Spatial distribution of normalized difference ($\rho$) for analytical test cases of MacDonald et al. (1995).





## 4.2 Synthetic test cases

Results for the baseline synthetic test, Case 1, are shown in Figure 14. The SPRNT-RS method produces visually the same solution to HEC-RAS with $z_R = z_0$. Similarly, the comparison of model results for depth ($h_0$) for test Cases $2-5$ are visually indistinguishable as shown in the left column of Figure 15. The quantitative difference measures for the synthetic tests are provided in Table 5 and the spatial distributions of $\rho(x)$ are illustrated in the right column of Fig. 15. Values for $\rho(x)$ in Cases 2 and 3 are slightly below zero ($\approx 0.02\%$) over the entire domain, indicating the SPRNT-RS solution has a slightly higher

water surface than the HEC-RAS solution for small perturbations in the bed slope. With the increased bottom perturbations in Cases 4 and 5 the $\rho(x)$ range is larger (and includes a positive range) but the bounding values are still trivial. The $\zeta$ and RMSE measures show that the non-dimensional and dimensional overall differences are small. The MAE and RMSE climb slightly with the increasing $h_R$ for Cases 2 through 5 but remains below 3 mm. These depth RMSE values are negligible compared with the normal depth (4.95 m) of the baseline and well within reasonable truncation error differences for solvers using different

numerical techniques. The model-model comparisons for test Cases A and B also have trivial errors (Table 5), and further results are not shown as they are visually identical to those for baseline Case 1 illustrated in Fig. 14. Note that the RMSE for both cases is identical to Case 1, which indicates solution for the SPRNT-RS method with $S_R \neq S_0$ is very close to the baseline solution with $S_R = S_0$.

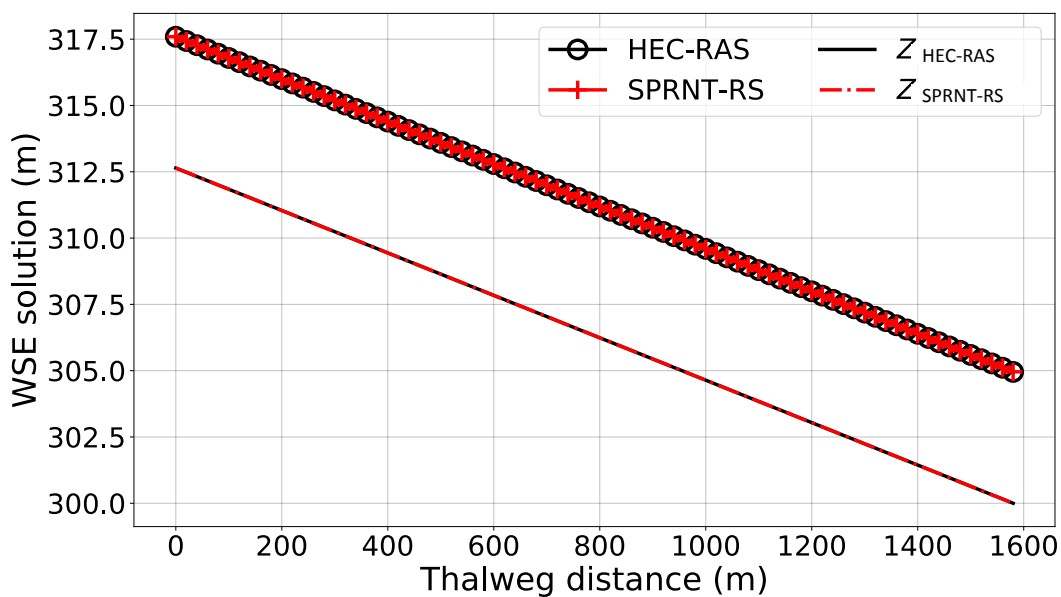

**Figure 14.** Simulated profile of water surface elevation (upper line) and channel bottom (lower line) for synthetic test Case 1 using SPRNT-RS (red), and unsteady HEC-RAS (black).





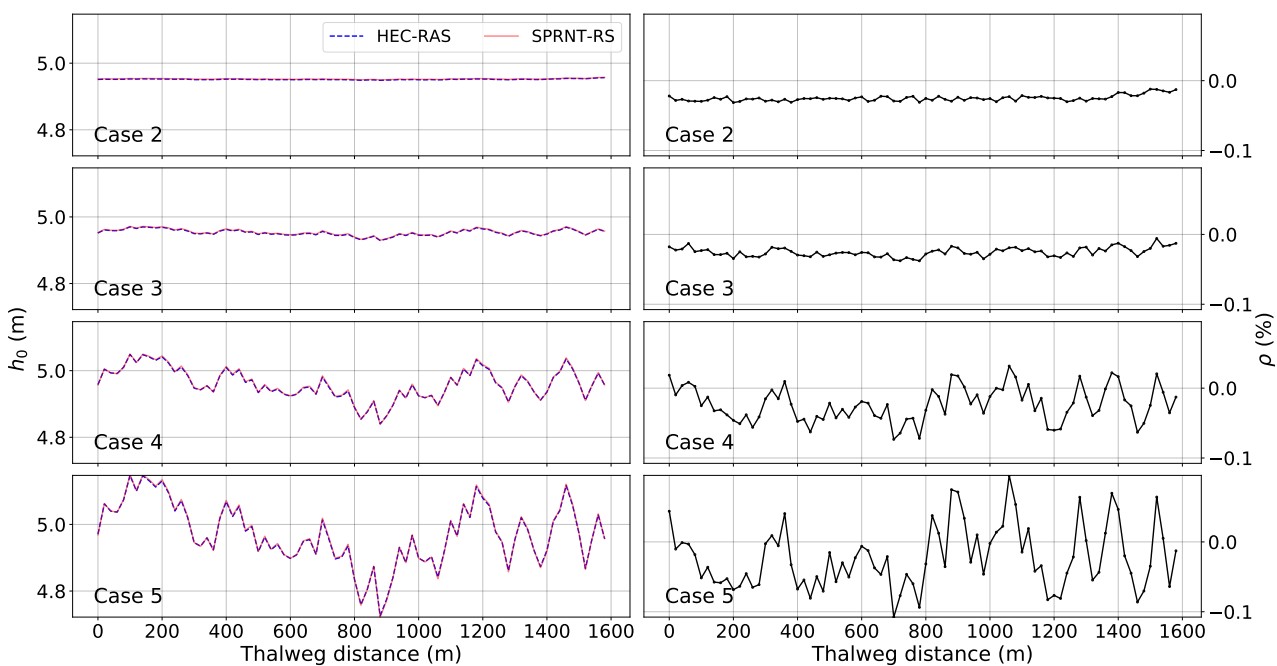

**Figure 15.** Water depth, $h_0$, (left column) and normalized difference, $\rho$, (right column) for synthetic test cases with perturbed bathymetry.

| Case | min($\rho$) | max($\rho$) | $\zeta$ | MAE (m) | RMSE (m) |
|---|---|---|---|---|---|
| Case 1 (baseline) | $-0.107\%$ | $0.005\%$ | $0.0281\%$ | $0.00139$ $(0.028\%)$ | $0.00176$ $(0.035\%)$ |
| Case 2 | $-0.031\%$ | $-0.012\%$ | $0.0251\%$ | $0.00124$ $(0.025\%)$ | $0.00126$ $(0.025\%)$ |
| Case 3 | $-0.037\%$ | $-0.005\%$ | $0.0252\%$ | $0.00125$ $(0.025\%)$ | $0.00129$ $(0.026\%)$ |
| Case 4 | $-0.073\%$ | $0.032\%$ | $0.0284\%$ | $0.00141$ $(0.028\%)$ | $0.00168$ $(0.034\%)$ |
| Case 5 | $-0.107\%$ | $0.095\%$ | $0.0426\%$ | $0.00212$ $(0.043\%)$ | $0.00249$ $(0.049\%)$ |
| Case A | $-0.107\%$ | $0.005\%$ | $0.0281\%$ | $0.00139$ $(0.028\%)$ | $0.00175$ $(0.035\%)$ |
| Case B | $-0.107\%$ | $0.005\%$ | $0.0281\%$ | $0.00139$ $(0.028\%)$ | $0.00175$ $(0.035\%)$ |

**Table 5.** Difference measures between SPRNT-RS and HEC-RAS using eqs. (23) – (28) for synthetic test cases. Non-dimensionalized MAE and RMSE are shown in parentheses.





### 4.3 Waller Creek test case

Waller Creek has been simulated with SPRNT-RS (denoted as $WC_{RS}$ in the following figures), the HEC-RAS unsteady solver ($WC_{HEC-U}$) and the HEC-RAS steady solver ($WC_{HEC-S}$). Figure 16 shows water surface elevations for SPRNT-RS and unsteady HEC-RAS. For clarity the upper 40% of the domain (which has similar good behavior) is not shown. Figure 17 shows the spatial distribution of the normalized difference $\rho$ for these simulations. The differences are roughly within $\pm 4\%$ across the entire domain. The maximum and minimum difference both occur at two adjacent nodes close to 7800 m with $4.14\%$ and

$-3.07\%$, respectively. Figure 18 provides a similar comparison of water surface elevations between SPRNT-RS and the steady HEC-RAS case. The results are visually quite similar to the comparison with unsteady HEC-RAS. A direct comparison of surface elevations for unsteady and steady HEC-RAS does not provide any further insight and is omitted for brevity. However, to quantitatively evaluate the differences between SPRNT-RS and HEC-RAS, it is useful to compute difference measures between the unsteady and steady HEC-RAS models themselves as well as the differences between SPRNT-RS and both models,

as provided in Table 6. Overall, the SPRNT-RS result have marginally better consistency with unsteady HEC-RAS than with steady HEC-RAS. Of greater importance is that the behavior of SPRNT-RS relative to unsteady HEC-RAS has the same order of differences as the comparison of unsteady HEC-RAS to steady HEC-RAS. These results imply that the differences between SPRNT-RS and unsteady HEC-RAS are reasonable for the different numerical methods given the geometric variability of Waller Creek.

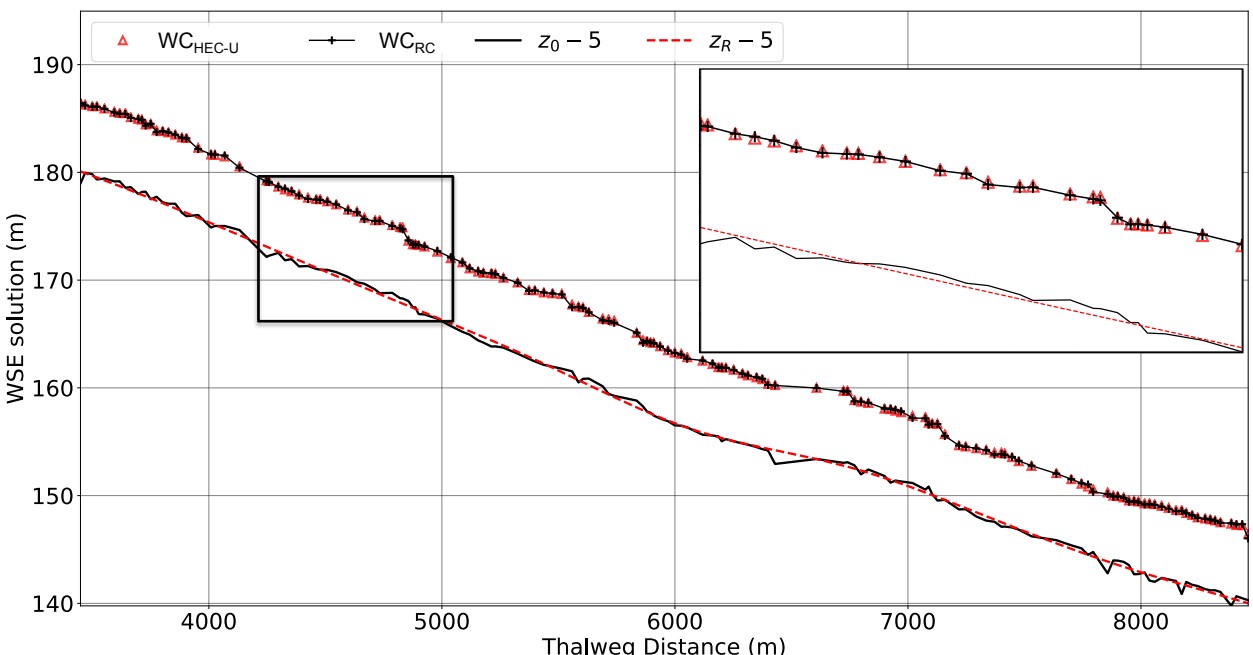

**Figure 16.** Comparison of SPRNT-RS to unsteady HEC-RAS for water surface elevations in Waller Creek simulations with expanded detail to show similarities. For clarity, 5 m is subtracted from the channel $z$ elevations.





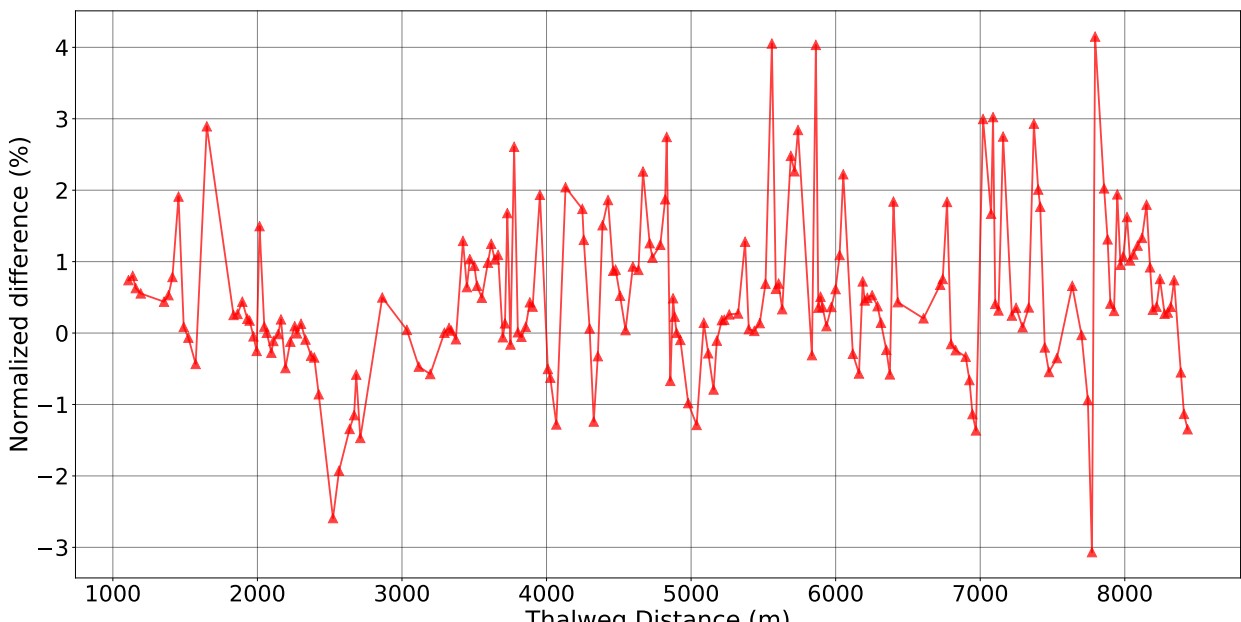

**Figure 17.** Normalized difference $\rho(x)$ between SPRNT-RS and unsteady HEC-RAS

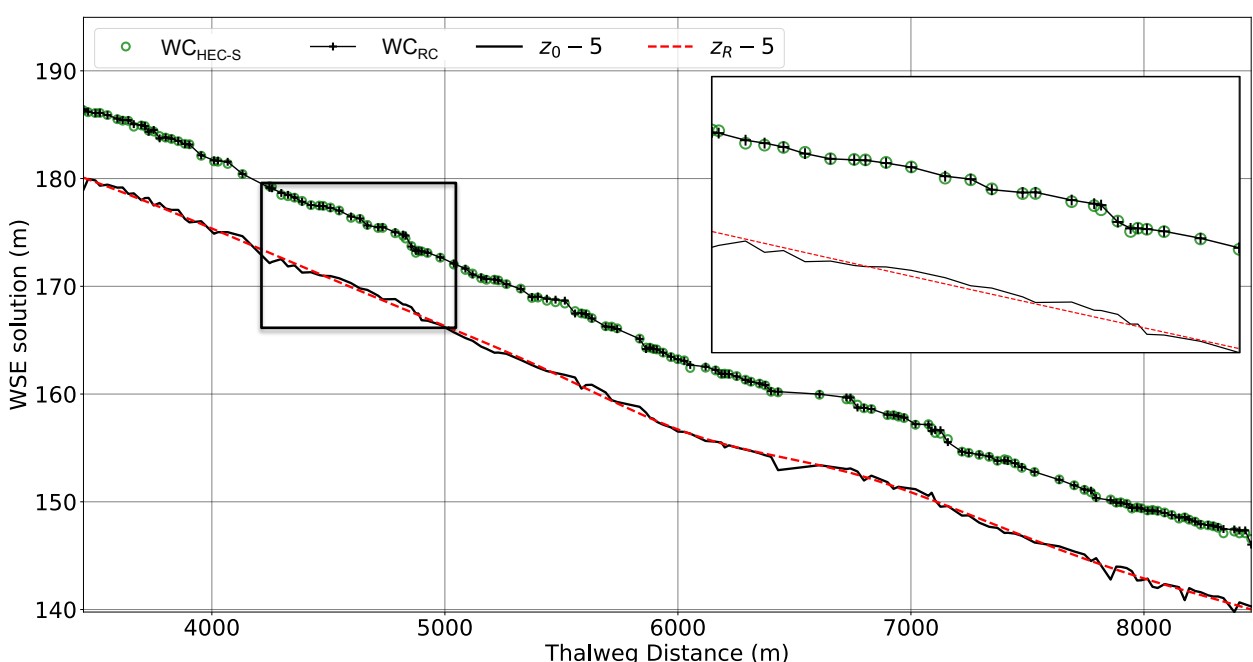

**Figure 18.** Comparison of SPRNT-RS to steady HEC-RAS for water surface elevations in Waller Creek simulations with expanded detail to show similarities. For clarity, 5 m is subtracted from the channel $z$ elevations.





| Case comparison | $\min(\rho)$ | $\max(\rho)$ | $\zeta$ | MAE (m) | RMSE (m) |
|---|---|---|---|---|---|
| $\text{WC}_{\text{RS}} : \text{WC}_{\text{HEC-U}}$ | -3.07% | 4.14% | 0.85% | 0.056 (0.883%) | 0.077 (1.284%) |
| $\text{WC}_{\text{RS}} : \text{WC}_{\text{HEC-S}}$ | -8.43% | 4.58% | 1.25% | 0.081 (1.287%) | 0.122 (1.990%) |
| $\text{WC}_{\text{HEC-U}} : \text{WC}_{\text{HEC-S}}$ | -2.70% | 7.03% | 1.30% | 0.086 (1.301%) | 0.128 (1.931%) |

**Table 6.** Difference metrics for Waller Creek simulation results. Non-dimensionalized MAE and RMSE are shown in parentheses.

## 5 Discussion

### 5.1 Validation of the RS method

The RS method is a simple algebraic transformation of the governing equations and the answer to the principal question "does it work?" is implied by our inability use the baseline SPRNT model (with $S_0$) as a control model (see §3.4). Invariably, discontinuous topography for SPRNT without RS caused either an oscillatory solution or numerical instability. In contrast, both HEC-RAS (using $\eta$) and SPRNT-RS (using $S_R$) provide stable, non-oscillatory solutions.

The analytical results in §4.1, supplemented by additional results in Yu et al. (2019), validate the SPRNT-RS method for simulation of smoothly-varying channel morphologies that are Lipschitz continuous at the discretization scale. We have experimented with both uniform and splined $S_R$ for these tests. For both types of simulations we observe errors relative to physical experiments that are comparable or smaller than those shown in the numerical validation studies of MacDonald et al. (1995). These results imply that the transformation from the conventional $h_0, S_0$ form of the SVE to the $h_a, S_R$ form of eq. (17) is a valid algebraic step that can be implemented in a numerical solver and is an alternative for representing smooth geometries.

The synthetic test cases in §4.2 serve two purposes. Firstly, Cases A and B compared to baseline Case 1 show that the numerical solution does not depend on a particular choice of $S_R$. Arbitrary selection of an $S_R \neq S_0$ results in identical solutions to $S_R = S_0$. Secondly, the synthetic test cases show that the SPRNT-RS method can be applied with non-smooth geometry at the discretization scale (i.e., random perturbations of the physical bottom slope), which caused non-convergent behavior in the baseline SPRNT model. As a control, we have compared SPRNT-RS with the accepted HEC-RAS model that remains stable for these test cases as it solves with the piezometric pressure gradient rather than splitting into $S_0$ and the gradient of $h_0$. The results indicate that SPRNT-RS provides numerical solutions that are nearly identical to HEC-RAS for the non-smooth geometry test cases. Thus, using a Lipschitz-smooth $S_R$ provides a stable numerical solution for non-smooth geometry *without altering the physical representation of non-smooth geometry*.

The Waller Creek test case in §4.3 provides a more challenging comparison of SPRNT-RS to HEC-RAS. For this test case, the geometry discontinuities include adverse slopes and local $S_0$ that are $\pm400\%$ of the reach-average slope, which contrasts with perturbations of $\pm30\%$ used in the synthetic test cases of §4.2. Again, SPRNT-RS is shown to be close to the unsteady HEC-RAS solution. The model differences are within reasonable ranges, as illustrated by the fact that they are similar to the differences between HEC-RAS steady and unsteady versions. Nevertheless, it remains possible that the minor differences





between HEC-RAS and SPRNT-RS are caused by a latent defect in coding the RS method or SPRNT itself, but it is difficult to envision how such a defect could occur without also appearing in the analytical and synthetic test cases. A simpler and more compelling explanation is with the linear approximations used in unsteady HEC-RAS that are not present in SPRNT-RS. Specifically, Brunner (2016a) notes that for computational efficiency and to reduce "troublesome convergence problems at

discontinuities in the river geometry," the unsteady HEC-RAS code uses a linearization technique developed by Liggett (1975) and Chen (1973) – note the latter document is cited by Brunner (2016a) but was not available to us. It seems likely that strong geometry discontinuities in the Waller Creek test case would be affected by this linearization, which arguably would lead to artificial smoothing of the water surface profile by HEC-RAS. Unfortunately, we do not have direct access to the HEC-RAS code and thus rely on the discussion of HEC-RAS stability in the literature (Hicks and Peacock, 2005; Sharkey, 2014) and the

methodology in HEC-RAS manuals (Brunner, 2016a, b).

### 5.2 Why not just use $\partial\eta/\partial x$?

One might wonder whether $S_R$ or $S_0$ is at all necessary when we could clearly just retain $\partial\eta/\partial x$ in the SVE rather than using any split form. To understand the value of $S_R$, it is worth considering why $S_0$ is presently used. We have not been able to determine exactly when $S_0$ was first used with the SVE, but from a hydrology viewpoint $S_0$ provides consistency between

kinematic wave solutions (which use $S_0 = S_f$) and the SVE. Thus including $S_0$ is a logical step when considering reduced-physics approaches (Di Baldassarre, 2012). Arguably, a well-chosen $S_R$ that matches the large-scale $S_0$ will serve the same purpose. The $S_0$ approach is also favored in models that are built on a "conservative" SVE form where the hydrostatic pressure portion of the piezometric head gradient is abstracted into the advective gradient term (e.g. Sanders, 2001; Kesserwani, 2013). For these model, the advantage of the $S_0$ form is that when $S_0 = 0$ and $S_f = 0$ the momentum equation can be written as a

classic 1D homogenous advection equation, which is mathematically appealing. Our work in progress indicates that the $S_R$ approach could be similarly adapted for a conservative form of the SVE, but this issue remains speculative.

Although the utility and simplicity of the $\eta$ approach is obvious, it has a key disadvantage when applied in large-scale simulations. Over large distances the free surface is monotonically increasing upstream, which has consequences for employing implicit or semi-implicit numerical solutions in a continental river dynamics framework (Hodges, 2013). Briefly, when mod-

eling a river system from an estuary ($\eta \sim 0$ m) to mountain headwaters ($\eta \sim 10^3$ m) the solution variable $\eta$ nominally covers three orders of magnitude. Furthermore, as local variations on the order of $10^{-2}$ m affect the hydrostatic pressure gradient, the solution of $\eta$ requires precision over at least five orders of magnitude – i.e., a stiff numerical solution that can be difficult to converge for either a linear or nonlinear solver. Thus, splitting $\partial\eta/\partial x$ into a down-slope body force ($S_R$) and a local residual ($\partial h_a/\partial x$) is effectively removing a large-scale gradient from the solution variables, which will generally improve numerical

behavior.

Despite the above disadvantages, the $\eta$ form retains some advantages in creating conservative finite-volume formulations of the Saint-Venant equations (Hodges, 2019). Arguably, such methods should be confined to explicit time-marching schemes or localized solutions where $\eta$ covers a smaller range, and the RS method should be preferred for larger systems.





### 5.3 RS advantages and limitations

The fundamental difference between the SPRNT-RS approach and most, if not all, conventional models (including unsteady HEC-RAS) is that our method algebraically revises the Saint-Venant equations to *exactly* accommodate discontinuous geometry while maintaining a smooth source term, whereas other models typically introduce *ad hoc* changes (e.g., linearization) to provide stable and faster numerical behavior when discontinuities are likely to cause numerical instabilities. These differences in the governing equations can be expected to cause differences in the solution – especially where nonlinear terms are strong.

An important limitation to the present work is that we focus solely on subcritical flow. The Preissmann scheme used in the underlying SPRNT model is known to exhibit instabilities with transcritical flows (Samuels and Skeels, 1990; Sart et al., 2010; Meselhe and Holly Jr, 1997), which can be suppressed with the *ad hoc* Local Partial Inertia (LPI) scheme of Fread et al. (1996). Our preliminary work (not shown) indicates that the RS approach can stabilize the Preissmann scheme without using LPI, but further work is required to test and validate the RS method for transcritical and supercritical flows.

Overall, the RS method can ensure the Lipschitz smoothness of slope representation in the momentum source term (without smoothing geometry), thus reducing one source of oscillatory or unstable behavior in numerical solutions. However, application of the RS method is not without some limitations. Although the switch from $S_0$ to $S_R$ is algebraically exact, the application of the RS method requires some method to select the distributed $z_R(x)$ and to determine $S_R(x)$. Poor selection of $z_R$ can theoretically result in non-smooth $S_R$. In the present work, a cubic B-spline technique used, which is controlled by the number 470 of "knots" and their spacing. In general, the distance between knots must be longer than the spacing between cross-sections so that the generated $S_R$ is smooth at the model's discretization scale. Much work remains to be done to establish optimum approaches for automatic generation of approximate splines for large river networks. We speculate that simple window filtering techniques may be adequate when river databases such as NHDplus.

This study has implemented RS only in the SPRNT code, as discussed in more detail in §3. The baseline governing equations 475 for SPRNT are of the form of eq. (2), the so-called "non-conservative" form – which simply means that the entirety of the hydrostatic pressure gradient is effectively a source term, as contrasted with "conservative" equations such as Cunge-Liggett form (Cunge et al., 1980), in which a portion of the hydrostatic pressure gradient is abstracted to the advection term. Although it remains to be shown in future work, the algebraic transformation implied in eq. (4) can arguably be applied in the Cunge-Liggett form or any other conservative form of the SVE. Similarly, the fundamental algebraic transformation to $S_R$ and $\partial h_a / \partial x$ 480 will be equally valid in any finite-volume method using $S_0$ and $h_0$.

The greatest barrier to adoption of the RS method in an existing model is likely the need to rewrite the geometry functions to accept $h_a$, $h_R$, and $z_R$ in place of $h_0$ and $z_0$. The difficulties involved in this effort depend on whether or not the model geometry functions are sufficiently isolated from the main solution algorithm. Indeed we can imagine codes where the geometry functions are essentially dispersed throughout and requires extensive effort to alter, debug, and verify.





## 5.4 The future for RS methods

The RS method as introduced above might be just a starting point. Although the present work focused on the non-conservative form, the concepts presented herein will likely be effective in addressing the "well-balanced" problem for conservative forms as reviewed in Kesserwani (2013) and Hodges (2019). Furthermore, the algebra in the RS demonstration above leads to the conjecture that the method could be extended to 2D reference slopes for bathymetry in 2D or 3D models. Undoubtedly there are unknown numerical challenges in extending to higher dimensions – particularly in ensuring a 2D spline function is adequately spaced to ensure smoothness – but there does not appear to be any fundamental conceptual difficulty in such efforts.

## 6 Conclusions

The reference slope (RS) method introduces a new form of the Saint-Venant equations for 1D river flow. The advantage of the RS method is that it ensures the body force (slope) source term is smooth and cannot destabilize the numerical solution. The RS method introduces the concept of an arbitrary smooth reference elevation, $z_R(x)$, with computed reference slopes, $S_R(x)$, and associated depths, $h_a(x)$. These geometries are algebraically related to the traditional channel thalweg elevation, depth, and bottom slope $(z_0, h_0, S_0)$ used in many models. The RS method is implemented in an open-source Saint-Venant solver as SPRNT-RS. In this study, SPRNT-RS was compared to both analytical solutions and the conventional HEC-RAS model for synthetic test reaches and an urban creek for subcritical flows. The model-model comparisons are within expected truncation error for the both analytical and synthetic test cases, and within acceptable differences for simulating flow through the complex geometry of an urban creek. The slightly larger simulation differences in the urban creek test case are likely due to *ad hoc* linearization algorithms used in HEC-RAS that do not appear in SPRNT-RS.

As discussed in the §2, when faced with non-smooth geometries in a channel reach, prior researchers have resorted to limiting or smoothing discontinuous source terms or employing numerical techniques that mitigate oscillatory/unstable numerical behaviors. In contrast, the new RS method transforms a discontinuous bottom slope source term into a smooth expression without losing either complexity in the geometry or introducing *ad hoc* smoothing of the geometry, the numerical method, or the solution. An important advantage of the RS method is that it is entirely mechanical – requiring only selection of control knot spacing for the approximating spline at some length scale larger than the cross-section spacing. That is, RS does *not* require the model designer or user to introduce smoothing thresholds or *ad hoc* algorithm bounds. As such, we believe the RS method could be particularly valuable as we move from from fine-resolution reach-scale modeling to large-scale continental river dynamics simulation (Hodges, 2013) or develop massively parallel stormwater network models for megacities (Morales-Hernandez et al., 2020)

The RS method is not specific to SPRNT, but can be adapted to any Saint-Venant solver that uses a bottom slope ($S_0$) term in the discretization. The mathematical change is conceptually trivial, but the actual effort depends on how cross-section geometry is embedded in the code. The code for both SPRNT and SPRNT-RS are available under open-source license at GitHub (Liu, 2014).





# 7 Code availability

Complete code for reference slope module and SPRNT are available at Github (https://github.com/frank-y-liu/SPRNT)

# 8 Data availability

All test case files and results are uploaded to a public repository under Texas Data Repository (https://doi.org/10.18738/T8/BXJBF5)

## Appendix A: Geometry adjustments for stable unsteady HEC-RAS simulations

As discussed in §3, the stability of the SPRNT-RS simulations for Waller Creek was ensured by merging 36 computational elements where the cross-section spacing was closer than 10 m. This minimum spacing cut-off was selected as being well

below the median spacing of 28.6 m and mean spacing of 37.7 and proved adequate for ensuring SPRNT-RS stability in the tested simulations. Unfortunately, stability of unsteady HEC-RAS required further removal of three cross-sectional elements (shown in Figure A1) and reducing Manning's $n$ at six additional cross-sections (listed in Table A1). Selecting these changes was a matter of art rather than science as we could not identify a clear criteria for cross-section removal or Manning's $n$ adjustment for HEC-RAS – other than these locations appeared to be where instabilities appeared in unsteady HEC-RAS

model runs. Although SPRNT-RS could run without these changes, for consistency in the model comparisons the geometry of the SPRNT-RS model was modified to exactly match the adjusted geometry required for the HEC-RAS model.

| station number | reach location (m) | original $n$ | modified $n$ |
|---|---|---|---|
| 30104 | 1384 | 0.06 | 0.04 |
| 30014 | 1412 | 0.06 | 0.04 |
| 29871 | 1454 | 0.055 | 0.04 |
| 29752 | 1490 | 0.06 | 0.04 |
| 29647 | 1522 | 0.055 | 0.04 |
| 29482 | 1572 | 0.055 | 0.04 |

**Table A1.** Modified Manning's $n$ for cross-section stations in Waller Creek data set to provide numerical stability of unsteady HEC-RAS.

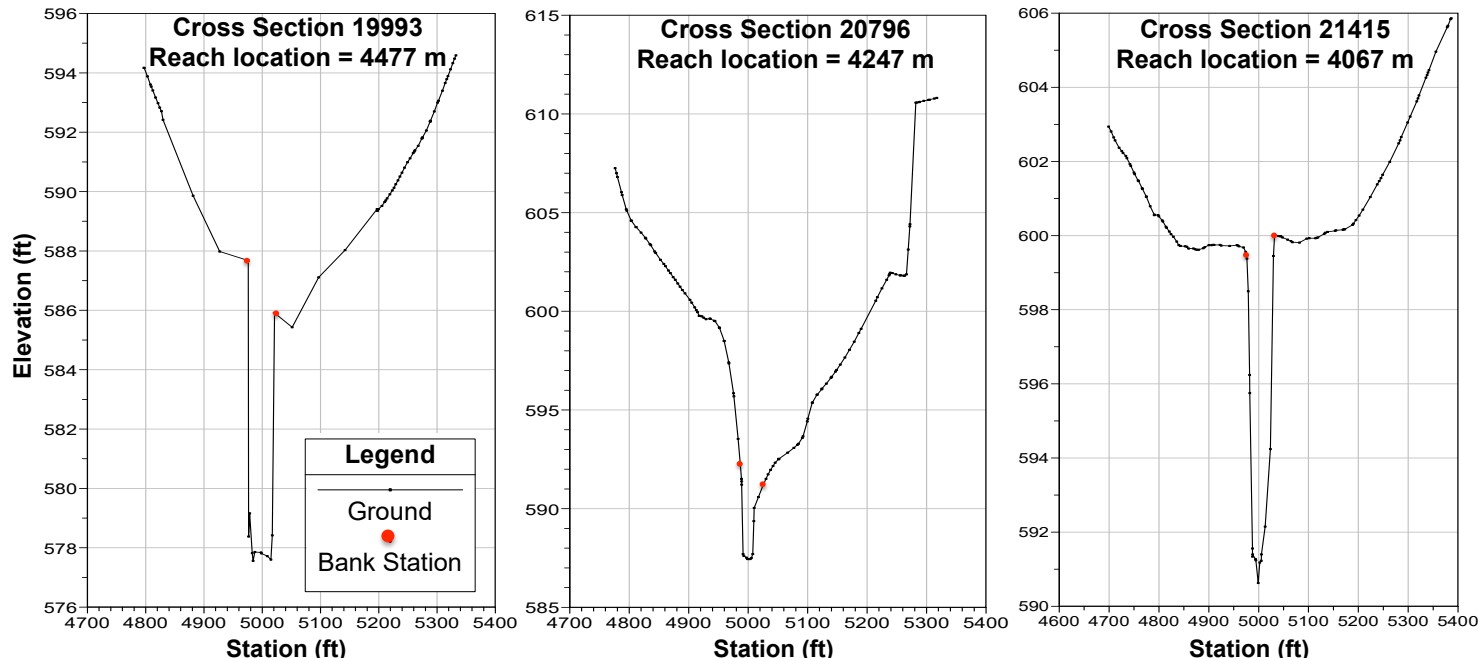

**Figure A1.** Cross-sections removed from Waller Creek data set to provide numerical stability of unsteady HEC-RAS.





*Author contributions.* C.-W. Yu and B. R. Hodges designed and performed the experiments, analysed the data, and wrote the manuscript. F. Liu wrote the code for the Reference Slope method.

*Competing interests.* The authors declare that they have no conflict of interest.

*Acknowledgements.* This research was supported by the U.S. National Science Foundation under grant number CCF-1331610. This article was also developed in part under Cooperative Agreement No. 83595001 awarded by the U.S. Environmental Protection Agency to The University of Texas at Austin. It has not been formally reviewed by EPA. The views expressed in this document are solely those of the authors and do not necessarily reflect those of the Agency. EPA does not endorse any products or commercial services mentioned in this publication. Authors would like to thank Dr. Fernando R. Salas from National Water Center, for providing asistant on collecting and processing the data

for test cases. The first author would also like to thank Alan Plummer Associates, Inc. for providing additional support during the latter stages of this project.

**Notation**

$A$ cross-sectional area (m$^2$)

$\beta$ momentum coefficient

$g$ gravitational acceleration (ms$^{-2}$)

$h_0$ water depth (m)

$h_a$ associated water depth (m)

$h_R$ reference height (m)

$n$ Manning's roughness (m$^{-1/3}$s)

$\eta$ water surface elevation (m)

$P_w$ wetted perimeter (m)

$Q$ volumetric flow rate (m$^3$s$^{-1}$)

$q_l$ flow rate per unit length through channel sides (m$^2$s$^{-1}$)

$\rho$ normalized difference between results

$S_0$ channel bottom slope

$S_f$ channel friction slope

$t$ time (s)

$v$ velocity (ms$^{-1}$)

$\bar{v}$ average velocity (ms$^{-1}$)

$W$ channel width (m)





$x$ along-channel spatial coordinate

$z_0$ channel bottom elevation (m)

$z_a$ reference elevation (m)

$\zeta$ absolute mean normalized difference (AMND)





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
