# Peer review of "A new form of the Saint-Venant equations for variable topography"

_Hydrology and Earth System Sciences, 2020_

## Referee Comment (RC1) · Anonymous Referee #1 · 27 Apr 2020

This work presents a interesting reformulation of the Saint-Venant equations in order to allow the inclusion of smoother geometrical source terms but maintaining a realistic representation of the arbitrary geometry of natural rivers and creeks. The proposed transformation splits the Piezometric gradient $\partial \eta / \partial x$ into a reference body force in the bottom slope direction $S_R$ and a hydrostatic head gradient $\partial h_a / \partial x$, ensuring that $S_R$ is Lipschitz continuous. The limit case for the proposed formulation is the widespread splitting form of the Saint-Venant equations, in which $S_R$ and $h_a$ agree with the thalweg slope $S_0$ and the maximum water depth $h_0$ at each section. The authors propose this simple algebraic transformation in order to avoid oscillatory solutions, or even unstable behavior, which the conventional splitting technique can cause in most of the numerical schemes when $S_0$ is Lipschitz discontinuous.

This work is original and well written. The tests carried out to demonstrate the applicability of the proposed technique are suitable and the discussion clear and well structured. From my point of view, I can see any important weakness in the mathematical approach and the discussion. Only some minor corrections must be included before this work can be considered for publication. Please, find the detailed comments in the attached pdf.

Please also note the supplement to this comment:
https://www.hydrol-earth-syst-sci-discuss.net/hess-2020-75/hess-2020-75-RC1-supplement.pdf

**Supplement:**

**Review for hess-2020-75**
**A new form of the Saint-Venant equations for variable topography**

This work presents a interesting reformulation of the Saint-Venant equations in order to allow the inclusion of smoother geometrical source terms but maintaining a realistic representation of the arbitrary geometry of natural rivers and creeks. The proposed transformation splits the Piezometric gradient $\partial \eta / \partial x$ into a reference body force in the bottom slope direction $S_R$ and a hydrostatic head gradient $\partial h_a / \partial x$, ensuring that $S_R$ is Lipschitz continuous. The limit case for the proposed formulation is the widespread splitting form of the Saint-Venant equations, in which $S_R$ and $h_a$ agree with the thalweg slope $S_0$ and the maximum water depth $h_0$ at each section. The authors propose this simple algebraic transformation in order to avoid oscillatory solutions, or even unstable behavior, which the conventional splitting technique can cause in most of the numerical schemes when $S_0$ is Lipschitz discontinuous.

This work is original and well written. The tests carried out to demonstrate the applicability of the proposed technique are suitable and the discussion clear and well structured. From my point of view, I can see any important weakness in the mathematical approach and the discussion. Only some minor corrections must be included before this work can be considered for publication.

My main concern is related to the validation methodology. The authors compare their numerical results with analytical solutions (MacDonald benchmarking cases) and with those obtained with HEC-RAS for synthetic cases and a urban creek case. HEC-RAS is widely-accepted model and uses Piezometric gradient version of the Saint-Venant equations in order to obtain stable solutions, avoiding numerical oscillation. As validation strategy, this comparison is correct and valuable. However, it would also be interesting to include the comparison with some of the existing well-balanced models based on the conventional splitting $\eta = z_0 + h_0$ and able to deal with discontinuous geometrical source terms. Although this reviewer understands that these models probably are not accessible for the authors, including such comparison for the urban creek case would increase the quality of the discussion. That is only a suggestion.

Other minor corrections:

Line 26: "...will be designated as 'reference slope', $S_R$,..."

Line 63: "...splitting of the Piezometric head to include a body force that

is everywhere aligned with a variable $S_0$ is merely creating an unnecessary complexity..."

The main advantage of including $S_0$ as a body force is that real disappointing in the topography, as chutes, are included into the forcing terms. Also, from a hydrology viewpoint, $S_0$ provides consistency between kinematic wave solutions (which use $S_0 = S_f$) and the SVE, as the author claim in Section 5.2. Hence this sentence should be explained in detail. Why including $S_0$ "is merely creating an unnecessary complexity"?

Line 97: "...Unfortunately, many water resources models do not use well-balanced schemes, and those that do are often computationally intensive and therefore impractical for simulating regional-to-continental scale river networks or stormwater systems for megacities..."

This sentence is misleading. Maybe can be reworded.

Line 127: "...even when $\partial A/\partial x$ is non-smooth..."

Line 189: "...Note that in extreme cases of geometric discontinuity the values of n, Pw and A in eq. (9) can cause a non-Lipschitz source term; however, most solution methods are relatively robust to such discontinuities as they are in the coefficient of the solution variable rather than an additive source term..."

Integration of friction source terms has been in main issue in numerical models during decades, specially when wet-dry fronts are involved. This led to a wide range of proposed solutions, from the implicit computation of the friction term to limiting its value for ensuring the positivity of the water depth solution. At least this should be mentioned in the text including some references.

Section 3.3 Generating a smooth $S_R(x)$: How the points of the real thalweg are selected to construct the reference profile $z_R$? Are there any optimization method to select them?

Figures 16 and 18: Line colors for the bed profile and the WSL are changed. Maybe it can be more appropriated that the bed and WSL lines have the same color for each model.

---

## Referee Comment (RC2) · Anonymous Referee #2 · 4 May 2020

The manuscript describes an original re-formulation of the Saint-Venant equations, based on the introduction of a properly defined, mathematically equivalent, modified slope term in the momentum equation. A new "reference slope" is introduced with the aim of removing the mathematical issues related to the lack of Lipschitz smoothness condition in the original source terms, which may represent a major source of spurious oscillations in the numerical solution. Several numerical tests of the proposed form of the Saint-Venant equations are performed, in order to check the consistency of the proposed procedure, and to benchmark the solution procedure against widespread standard methods. Pros and Cons of the proposed method are finally discussed in detail. The manuscript concerns a very interesting topic, proposing an original and innovative approach with some potential to trigger new approaches even in different

fields. The research is clear in its design and the manuscript is well-organised and generally well-written. I suggest addressing few minor issues, that could contribute to clarify at some points the manuscript and to improve the manuscript quality:

- Line 42. In commenting eq. (4) the Authors should explain what they intend for "associated depth consistent with the above definition"

- Line 185. The manuscript should better explain why the interest is to "non-trivial definitions of  $z_R(x)$  that are close to  $S_0(x)$  but are guaranteed smooth". This may appear to be in contrast with the circumstance that the mathematical re-formulation of the momentum equation is equivalent for any choice (close or not) of  $z_R(x)$  (for the purposes addressed in the paper, at least for any choice providing Lipschitz smoothness).

- Line 210. The discussion here ("approximating cubic B-splines to the  $z_0(x)$ ...") may suggest the idea that some choice of  $z_R(x)$  could be better than other ones. I think that a comment is needed.

- Line 471. The Authors should add in the revised manuscript a comment on the potential benefit provided by "automatic generation of approximate splines for large river network": is that essential for the successful application of the proposed methodology? Why?

- Please proofread the manuscript. For instance: Line 65. "an" should read "a". Line 473. Broken sentence? Line 266. "Being" should read "begin".

- Please double check the notation list. For example, momentum coefficient is in the notation list but not in the equations. Similarly, velocity and average velocity. Reference slope is not in the equation. Reference Slope is not in the notation list.

Based on the above comments, I am confident that the manuscript could be published, provided these few minor issues are accounted for in the revised version.

75, 2020.

---

## Author Comment (AC1) · 5 May 2020

Dear reviewer,

We appreciate your thorough review of the manuscript and all the constructive suggestions you provided. We will provide a complete response with a revised manuscript to address all the comments as soon as possible.

Besides, we want to make sure that we don't misinterpret the sentence. In the second paragraph of your comment, you mentioned: "I can see any important weakness in the mathematical approach and the discussion". We suppose it is a typo and the original sentence you want to write is: "I can't see any important weakness in the mathematical approach and the discussion".

[Figure]

Again, we are grateful for the review and will provide the complete response shortly.

Bests,

---

## Author Comment (AC2) · 5 May 2020

Dear reviewer,

We appreciate your comments and suggestions. The complete response will be provided shortly.

Bests,
* * *

---

## Referee Comment (RC3) · Anonymous Referee #1 · 10 May 2020

Dear author, you are right. It was a typo error. Best regards.

---

## Author Comment (AC3) · 29 May 2020

Dear reviewer, We herein provide a supplementary file to answer your comments and suggestions. In the file, the response and revised text in the manuscript are sequentially listed following the order of your comments. Please feel free to let us know if you have any further questions. We sincerely appreciate your review and help.

Bests,

Please also note the supplement to this comment: https://www.hydrol-earth-syst-sci-discuss.net/hess-2020-75/hess-2020-75-AC3-supplement.pdf

[Figure]

**Supplement:**

We thank reviewers for their thorough review of the manuscript and the constructive suggestions. The comments and suggestions are answered sequentially following the order of reviewers' questions and comments.

**Response for HESS-2020-75 Reviewer #1**
**A new form of the Saint-Venant equations for variable topography**

R1C1
*My main concern is related to the validation methodology. The authors compare their numerical results with analytical solutions (MacDonald benchmarking cases) and with those obtained with HEC-RAS for synthetic cases and a urban creek case. HEC-RAS is widely-accepted model and uses Piezometric gradient version of the Saint-Venant equations in order to obtain stable solutions, avoiding numerical oscillation. As validation strategy, this comparison is correct and valuable. However, it would also be interesting to include the comparison with some of the existing well-balanced models based on the conventional splitting $\eta = z_0 + h_0$ and able to deal with discontinuous geometrical source terms. Although this reviewer understands that these models probably are not accessible for the authors, including such comparison for the urban creek case would increase the quality of the discussion. That is only a suggestion.*

Response to R1C1:
**We agree with the idea but we cannot fully implement in the paper. We have added the discussion in Section 5.3 as shown below.**

> *Beginning Line 488: The present scope is limited in that only a single model was modified (SPRNT/SPRNT-RS) and only single model (HEC-RAS) was used as an external control. The validity of the underlying algebraic transformation in the RS method has been demonstrated by these tests; however, it remains to be seen how implementing the RS approach in other models -- particularly well-balanced models -- might alter residual errors, convergence rates, and computational performance. We are interested in collaborating with other researchers who have access to and familiarity with source code of candidate well-balanced models.*

For further clarification -- we appreciate reviewer's suggestion of using well-balanced model to explore how the RS method affects other codes, which we agree would be quite interesting and informative. However, this would not significantly add to the validation of the RS method – which is the focus of the present work (particularly since the key point is that the method itself is a simple algebraic transformation). Nevertheless, we are quite interested in how implementation of the RS method would alter the numerical performance of a well-balanced model. We hypothesize that it would perform better – but this is pure speculation. In any case, such an effort is an extension that is well beyond the present scope of work. Including further modeling would increase the length of the paper and dilute its focus. However, we do think it worth pointing out this limitation more clearly in the discussion, which we have done in the revised text.

R1C2
*Line 26: "...will be designated as 'reference slope', $S_{R}$,..."*

Response to R1C2:
**Agree. The sentence has been corrected.** We appreciate reviewer's carefulness.

R1C3

*Line 63: "...splitting of the Piezometric head to include a body force that is everywhere aligned with a variable $S_0$ is merely creating an unnecessary complexity..."*

*The main advantage of including $S_0$ as a body force is that real disappointing in the topography, as chutes, are included into the forcing terms. Also, from a hydrology viewpoint, $S_0$ provides consistency between kinematic wave solutions (which use $S_0 = S_f$) and the SVE, as the author claim in Section 5.2. Hence this sentence should be explained in detail. Why including $S_0$ " is merely creating an unnecessary complexity"?*

Response to R1C3:

**Agree. We have rewritten the explanation as noted below.**

*New text beginning Line 59:* From a physics perspective, using $S_0$ to split the Piezometric head is an intuitive way to describe the local interplay of pressure with the bottom slope. Furthermore, $S_0$ has the advantage of readily reducing to a kinematic wave equation where $S_f=S_0$, which has some advantage in multi-purpose codes. However from a numerical modeling perspective, using $S_0$ has a significant limitation based on its smoothness. If the water surface is smooth then non-smooth $S_0(x)$ requires the numerical solver to produce a compensating non-smooth $h_0(x)$, i.e., requiring a "well-balanced" scheme (see §2). If we can discard our (wrong) intuition that the $S_0$ form must somehow "better" represent sharply variable topography – i.e., recognizing the algebraic equivalence of eq. (5) with eqs. (1) and (2) – it follows that splitting of the Piezometric head to include a body force that is everywhere exactly aligned with a sharply varying $S_0$ is (from a numerical perspective) merely creating unnecessary complexity in the governing equation source term that requires compensating complexity in the solution algorithm. In contrast, by requiring $S_R$ to be smooth we can ensure the $h_a$ solution is also smooth for a smoothly-varying free surface.

For further clarification – note that real topography and its forcing is included in *any* Saint-Venant solution (using $S_0$, $S_R$, or the free surface) as long as the physical A(x) and the net piezometric pressure gradients are correctly identified by the governing equations. This is the key point of the algebraic equivalence of eq. (1), (2) and (5). If the equations are algebraically identical, they must represent the exact same physics. Hence S0 is not needed to represent the topography.

R1C4

*Line 97: "...Unfortunately, many water resources models do not use well-balanced schemes, and those that do are often computationally intensive and therefore impractical for simulating regional-to-continental scale river networks or stormwater systems for megacities..." This sentence is misleading. Maybe can be reworded.*

Response to R1C4:

**Agree: We have rewritten the explanation as noted below.**

*Beginning Line 102:* *Although well-balanced schemes are relatively robust in handling the discontinuous boundary conditions, they have not been extensively applied in water resources models to simulate the regional-to-continental scale river networks or stormwater systems for megacities. The rapidly varying and discontinuous S0 in natural systems can significantly increase the difficulty and computational burden of obtaining a well-balanced method (Schippa and Pavan, 2008). Hence, when a large-scale open-channel model develops oscillations and/or instabilities,*

*practitioners may resort to the traditional approach of removing cross-sections105or smoothing bathymetry to mitigate oscillatory or unstable solution behavior (Tayfur et al., 1993).*

R1C5
*Line 127: "...even when ∂A/∂x is non-smooth..."*
*Line 189: "...Note that in extreme cases of geometric discontinuity the values of n, $P_w$ and A in eq. (9) can cause a non-Lipschitz source term; however, most solution methods are relatively robust to such discontinuities as they are in the coefficient of the solution variable rather than an additive source term..."*

*Integration of friction source terms has been in main issue in numerical models during decades, especially when wet-dry fronts are involved. This led to a wide range of proposed solutions, from the implicit computation of the friction term to limiting its value for ensuring the positivity of the water depth solution. At least this should be mentioned in the text including some references.*

Response to R1C5:
**Agree: We have added the text below to the Background.**

> *Beginning line 114: That numerical instabilities are often caused by non-smooth source terms is not a new observation. A wide variety of numerical schemes have been developed to address this issue, including (e.g.) extensive work on wetting/drying (Liang and Marche, 2009; Song et al., 2011), positivity-preserving methods for coupled models (Singh et al., 2015) and implicit schemes that address stiffness of the nonlinear friction term (Xia and Liang, 2018). The literature in this area is vast – particularly if both 1D and 2D models are considered. For the present purposes we focus on only one part of the source term, S0, whose non-smoothness has previously been treated as a problem to be handled rather than as a problem that can be directly eliminated in the governing equations. Existing well-balanced schemes (see reviews noted above) seek to compensate for non-smoothness of all parts of the source term in the structure of the numerical discretization. Arguably, if the slope term is guaranteed smooth then a well-balanced scheme should be simpler to create.*

**We have also revised the Methods with the following clarification**

> Beginning line 208: Note that in extreme cases of geometric discontinuity the combined values of n, $P_w$ and A in eq. (9) can cause a non-Lipschitz friction term; thus, the RS method cannot guarantee that the entire source term is smooth. Numerical solution methods are usually robust to discontinuities in n and $P_w$ as they are coefficients of the solution variables {A,Q}. More subtle problems might arise due to discontinuities developed in the $Q^2A^{-10/3}$ ratio in eq. (9); countering incipient instabilities from this term requires other numerical strategies (e.g., Xia and Liang, 2018)

Note that Line 127 (now line 142) is in the Background whereas Line 189 (now line 208) is in the Methods. We chose to add the bulk of the revised text as a new discussion in the Background to address past work beyond the bottom slope. Note that a full review of this issue would be a major undertaking and is beyond the present scope of validating that we have a new method that works. Just as a matter of interest -- at last count Greenberg & Leroux 1996 paper on well-balanced methods has been cited 443 times!

R1C6
*Section 3.3 Generating a smooth $S_R(x)$: How the points of the real thalweg are selected to construct the reference profile $z_R$? Are there any optimization method to select them?*

Response to R1C6:
**This question points out some clarifications needed. We have added the text noted below to the Methods section**.

*Beginning Line 242:  It follows that there is some (limited) choice in the selection of the subset of $z_0(x)$ used as the spline knots, with different sets producing slightly different $\{z_R(x),h_R(x)\}$ over the domain. Each set is algebraically identical to the underling geometry so the generated solutions should be identical within truncation error. Implications of the method chosen for generating $z_R(x)$ are discussed in §5.3. Further details and test cases are provided in Yu et al. (2019b)*

**Additional clarifications provided in the Discussion section:**

*Beginning Line 503: In the present work, the profile of $z_R$ in the Waller Creek case is generated by the cubic B-spline technique, which is controlled by the number of "knots" and their spacing. In general, the distance between knots must be longer than the spacing between cross-sections so that the generated $S_R$ is smooth at the model's discretization scale.  It is  not clear that a mathematical "optimum" for selection of knots will necessarily exist, but there are likely (unknown) practical limits on knot selection spacing for "adequate" smoothness of $z_R(x)$. Our results indicate that approximating cubic B-splines are adequate for producing smooth $z_R$ for the tested geometries, and the solutions are robust to the selection of $z_R$ as long as $S_R$ is smooth (Yu et al., 2019b). However, it is likely there are limitations to applying the RS method in large-scale river network simulation that will make it difficult to use a simple globally-applied knot spacing. Such networks might consist of $10^4$ to $10^5$ reaches  spanning wide geographical regions with a variety of topology and inconsistent data availability. Some reaches may have well-defined cross-sections at close spacing, other reaches might be poorly documented (Hodges, 2013). Thus, it seems likely that a method for automatically generating approximating splines (or some other form of smoothing) would be useful, but such an advance arguably requires a method for quantitatively evaluating the "goodness" of a particular set of $z_R(x)$, which remains an open question. We speculate that simple window filtering techniques may be adequate for river databases such as NHDplus, but further investigation and examination are needed to better understand the interplay between the smoothing scales and the numerical solution using the RS method for large networks.*

R1C7
*Figures 16 and 18: Line colors for the bed profile and the WSL are changed. Maybe it can be more appropriated that the bed and WSL lines have the same color for each model.*

Response to R1C7:
**Agree. The colors of the lines in Figure 16 and 18 have been made consistent.**

---

## Author Comment (AC4) · 29 May 2020

Dear reviewer, We herein provide a supplementary file to answer your comments and suggestions. In the file, the response and revised text in the manuscript are sequentially listed following the order of your comments. Please feel free to let us know if you have any further questions. We sincerely appreciate your review and help.

Bests,

Please also note the supplement to this comment:
https://www.hydrol-earth-syst-sci-discuss.net/hess-2020-75/hess-2020-75-AC4-supplement.pdf

[Figure]

**Supplement:**

We thank reviewers for their thorough review of the manuscript and the constructive suggestions. The comments and suggestions are answered sequentially following the order of reviewers' questions and comments.

**Response for HESS-2020-75 Reviewer #2**
**A new form of the Saint-Venant equations for variable topography**

R2C1
Line 42. In commenting eq. (4) the Authors should explain what they intend for "associated depth consistent with the above definition"

Response to R2C1:
**Agree. We have modified the text as shown below**

> *Beginning Line 42: where $S_R$ is an arbitrary reference slope and ha is an "associated depth" that will defined in §3.2, below. For an introductory exposition, $\partial ha/\partial x$ is merely the residual implied for a given $\partial \eta/\partial x$ and arbitrary $S_R$. Applying eq. (4) to eq. (1) provides:*

Note that in this introductory section we are trying to make things clear without providing the full details that are in the Methods

R2C2
Line 185. The manuscript should better explain why the interest is to "non-trivial definitions of $z_R(x)$ that are close to $S_0(x)$ but are guaranteed smooth". This may appear to be in contrast with the circumstance that the mathematical re-formulation of the momentum equation is equivalent for any choice (close or not) of $z_R(x)$ (for the purposes addressed in the paper, at least for any choice providing Lipschitz smoothness).

Response to R2C2:
**Agree. We have modified the text as shown below.**
Beginning Line 199: However, this form with $\partial \eta/\partial x$ for the entire pressure term is known to cause numerical stiffness issues for large ranges in η; e.g., the elevation change of a river from its mountain source to a coastal plain (Liu and Hodges, 2014). Using the conventional $S_0$ in eq. (2) reduces this problem as the range of $h_0$ is inherently confined to the local water depths rather than the underlying topography. In the RS method, the range of $h_a$ is tied to the range of water depths and the selection of $z_R$; thus, for present purposes we are interested in non-trivial definitions of $z_R$ that are (i) close to $z_0$ to maintain a small range of $h_a$ values, and (ii) provide smooth $S_R(x)$. Arbitrary $z_R$ that are far from $z_0$ or non-smooth are of little interest as they hold no theoretical or practical advantage over the eq. (1) approach implied by $S_R(x) = 0$.

R2C3
Line 210. The discussion here ("approximating cubic B-splines to the $z_0(x)$. . .") may suggest the idea that some choice of $z_R(x)$ could be better than other ones. I think that a comment is needed.

Response to R2C3:

**Agree. This is the same point as R1C6, answered above.**

> See new text beginning Line 242 and new text beginning line 503 as provided in answer to R1C6, above.

R2C4
Line 471. The Authors should add in the revised manuscript a comment on the potential benefit provided by "automatic generation of approximate splines for large river network": is that essential for the successful application of the proposed methodology? Why?

Response to R2C4:
**Agree: We have modified the discussion of splines in two places in response to R1C6 and R2C3. These revisions include the discussion noted here. The following text specifically addresses this point**

> *Beginning line 506:* *Our results indicate that approximating cubic B-splines are adequate for producing smooth $z_R$ for the tested geometries, and the solutions are robust to the selection of $z_R$ as long as $S_R$ is smooth (Yu et al., 2019b). However, it is likely there are limitations to applying the RS method in large-scale river network simulation that will make it difficult to use a simple globally-applied knot spacing. Such networks might consist of $10^4$ to $10^5$ reaches spanning wide geographical regions with a variety of topology and inconsistent data availability. Some reaches may have well-defined cross-sections at close spacing, other reaches might be poorly documented (Hodges, 2013). Thus, it seems likely that a method for automatically generating approximating splines (or some other form of smoothing) would be useful, but such an advance arguably requires a method for quantitatively evaluating the "goodness" of a particular set of $z_R(x)$, which remains an open question. We speculate that simple window filtering techniques may be adequate for river databases such as NHDplus, but further investigation and examination are needed to better understand the interplay between the smoothing scales and the numerical solution using the RS method for large networks.*

R2C5
Please proofread the manuscript. For instance: Line 65. "an" should read "a". Line 473. Broken sentence? Line 266. "Being" should read "begin".

Response to R2C5:
**Agree. Proofreading has been done. We've corrected the addressed places and we appreciate reviewer's carefulness.**

R2C6
Please double check the notation list. For example, momentum coefficient is in the notation list but not in the equations. Similarly, velocity and average velocity. Reference slope is not in the equation. Reference Slope is not in the notation list.

Response to R2C6:
**Agree. We've edited the notation list by adding/removing the missing/unnecessary notations. We appreciate reviewer's thorough review.**

---

## Author Comment (AC5) · 5 Jun 2020

Thank you for the clarification. The final response has beeb posted.
* * *